# R7 photoreceptor axon targeting depends on the relative levels of *lost and found* expression in R7 and its synaptic partners

Jessica Douthit[1†], Ariel Hairston[1†], Gina Lee[1], Carolyn A Morrison[1], Isabel Holguera[2], Jessica E Treisman[1]*

[1]Kimmel Center for Biology and Medicine at the Skirball Institute and Department of Cell Biology, NYU School of Medicine, New York, United States; [2]Department of Biology, New York University, New York, United States

**Abstract** As neural circuits form, growing processes select the correct synaptic partners through interactions between cell surface proteins. The presence of such proteins on two neuronal processes may lead to either adhesion or repulsion; however, the consequences of mismatched expression have rarely been explored. Here, we show that the *Drosophila* CUB-LDL protein Lost and found (Loaf) is required in the UV-sensitive R7 photoreceptor for normal axon targeting only when Loaf is also present in its synaptic partners. Although targeting occurs normally in *loaf* mutant animals, removing *loaf* from photoreceptors or expressing it in their postsynaptic neurons Tm5a/b or Dm9 in a *loaf* mutant causes mistargeting of R7 axons. Loaf localizes primarily to intracellular vesicles including endosomes. We propose that Loaf regulates the trafficking or function of one or more cell surface proteins, and an excess of these proteins on the synaptic partners of R7 prevents the formation of stable connections.

**\*For correspondence:**
Jessica.Treisman@med.nyu.edu

[†]These authors contributed equally to this work

**Competing interests:** The authors declare that no competing interests exist.

## Introduction

During nervous system development, growing axons must navigate through a complex environment and select the correct synaptic partners from numerous potential choices. Recognition of cell surface molecules plays an important role in axon guidance and targeting and the establishment of specific synaptic connections (*Yogev and Shen, 2014*). Interactions between cell surface molecules can lead to either adhesion or repulsion, and their relative levels on different cells are important for appropriate connections to form. For instance, gradients of ephrins and their Eph receptors enable retinal axons to form a topographic map in visual areas of the brain because Eph levels determine the sensitivity to ephrins (*Triplett and Feldheim, 2012*). In the *Drosophila* olfactory system, olfactory receptor neurons preferentially connect to projection neurons that express matching levels of the adhesion molecule Teneurin (*Hong et al., 2012*). As defects in synaptic adhesion molecules can lead to autism and other neurodevelopmental disorders (*Van Battum et al., 2015*; *Gilbert and Man, 2017*), identifying mechanisms that regulate synaptic partner choice is likely to enhance our understanding of such human diseases.

The *Drosophila* visual system has been a fruitful model for investigations of circuit assembly and synaptic specificity (*Plazaola-Sasieta et al., 2017*). The two color photoreceptors in the fly retina, R7 and R8, project to distinct layers in the medulla, M6 and M3 respectively. The R7 growth cone first actively targets a temporary layer, and then passively reaches its final layer due to the growth of other neuronal processes (*Ting et al., 2005*; *Özel et al., 2015*). Early stabilization of the R7 and R8 growth cones in different layers depends on differences in their relative levels of the transcription

**eLife digest** New nerve cells in a developing organism face a difficult challenge: finding the right partners to connect with in order to form the complex neural networks characteristic of a fully formed brain. Each cell encounters many potential matches but it chooses to connect to only a few, partly based on the proteins that decorate the surface of both cells. Still, too many cell types exist for each to have its own unique protein label, suggesting that nerve cells may also use the amount of each protein to identify suitable partners.

Douthit, Hairston et al. explored this possibility in developing fruit flies, focusing on how R7 photoreceptor cells – present in the eye to detect UV light – connect to nerve cells in a specific brain layer. It is easy to spot when the process goes awry, as the incorrect connections will be in a different layer. Experiments allowed Douthit, Hairston et al. to identify a protein baptized 'Lost and found' – 'Loaf' for short – which R7 photoreceptors use to find their partners.

Removing Loaf from the photoreceptors prevented them from connecting with their normal partners. Surprisingly though, removing Loaf from both the eye and the brain solved this problem – the cells, once again, formed the right connections. This suggests that R7 photoreceptors identify their partners by looking for cells that have less Loaf than they do: removing Loaf only from the photoreceptors disrupts this balance, leaving the cells unable to find their match. Another unexpected discovery was that Loaf is not present on the surface of cells, but instead occupies internal structures involved in protein transport. It may therefore work indirectly by controlling the movement of proteins to the cell surface.

These findings provide a new way of thinking about how nerve cells connect. In the future, this may help to understand the origins of conditions in which the brain is wired differently, such as schizophrenia and autism.

factor Sequoia (Seq); the adhesion molecule N-cadherin (Ncad) is thought to be the relevant target of Seq in these cells (*Petrovic and Hummel, 2008*; *Kulkarni et al., 2016*). Both Ncad and the receptor protein tyrosine phosphatase (RPTP) Lar are required to stabilize R7 terminals in the M6 layer. In the absence of either protein they remain in the M3 layer, although defects are observed earlier in development in *Ncad* mutants than in *Lar* mutants (*Clandinin et al., 2001*; *Lee et al., 2001*; *Maurel-Zaffran et al., 2001*; *Ting et al., 2005*; *Özel et al., 2015*; *Özel et al., 2019*). Another RPTP, Ptp69D, is partially redundant with Lar, and the depth of R7 axon termination correlates with the total level of RPTP activity (*Newsome et al., 2000*; *Hofmeyer and Treisman, 2009*; *Hakeda-Suzuki et al., 2017*). Stabilization of R7 contacts also requires the presynaptic proteins Liprin-α and Syd-1 that act downstream of Lar (*Choe et al., 2006*; *Hofmeyer et al., 2006*; *Holbrook et al., 2012*; *Özel et al., 2019*).

The primary synaptic targets of R7 that are responsible for its function in driving the spectral preference for ultraviolet light are the Dm8 medulla interneurons (*Gao et al., 2008*; *Takemura et al., 2013*; *Karuppudurai et al., 2014*; *Ting et al., 2014*). These cells fall into two subclasses, yellow (y) and pale (p), and their survival depends on their correct pairing with the appropriate R7 cell subtype, expressing either Rh4 (yR7) or Rh3 (pR7) (*Courgeon and Desplan, 2019*; *Menon et al., 2019*). The synapses R7 cells form on Dm8 cells often include the projection neurons Tm5a (for yR7s) or Tm5b (for pR7s) as a second postsynaptic element (*Gao et al., 2008*; *Takemura et al., 2013*; *Menon et al., 2019*). Another interneuron, Dm9, is both pre- and postsynaptic to R7 and R8 and mediates inhibitory interactions between ommatidia (*Takemura et al., 2013*; *Takemura et al., 2015*; *Heath et al., 2020*). It is not known which, if any, of these cell types provide Ncad or RPTP ligands that stabilize filopodia from the R7 growth cone (*Yonekura et al., 2007*; *Hofmeyer and Treisman, 2009*; *Hakeda-Suzuki et al., 2017*; *Özel et al., 2019*). Glia are also involved in establishing the pattern of R7 synaptogenesis, as they prevent excessive synapse formation through the adhesion protein Klingon (Klg) and its partner cDIP (*Shimozono et al., 2019*).

Here we identify a novel CUB-LDL domain transmembrane protein, Lost and found (Loaf), that acts in photoreceptors to promote the formation of stable R7 contacts in the M6 layer. R7 mistargeting to the M3 layer is observed when *loaf* function is lost from photoreceptors, but not in a fully *loaf* mutant animal. Similar defects can be induced in *loaf* mutants by expressing Loaf in neurons that

include Tm5a, Tm5b, Dm9, and Dm8, suggesting that R7 targeting is disrupted when Loaf is absent from R7 but present in its postsynaptic partners. Loaf does not itself promote cell adhesion and localizes primarily to endosomes. We propose that Loaf controls the trafficking or function of cell surface molecules that are used to match R7 to the correct postsynaptic neurons.

## Results

### *lost and found* is required in photoreceptors for normal R7 axon targeting

A microarray-based screen for genes with enriched expression in the R7 and R8 photoreceptors relative to R1-R6 identified *CG6024*, which encodes an uncharacterized transmembrane protein (*Pappu et al., 2011*). *CG6024* is also a predicted target of Glass (*Naval-Sanchez et al., 2013*), a transcription factor required for photoreceptor differentiation and axon guidance (*Moses et al., 1989*; *Selleck and Steller, 1991*). To test whether CG6024 has a function in axon targeting by R7 or R8, we expressed RNAi transgenes targeting *CG6024* with two different drivers: *GMR-GAL4* drives expression in all differentiating cell types in the eye (*Freeman, 1996*), and removing a stop cassette from *Actin>CD2>GAL4* with the eye-specific recombinase *ey*[3.5]*-FLP* (*Bazigou et al., 2007*) leads to RNAi expression in the entire eye disc. In both cases, R8 targeting was unaffected, but we observed a loss of R7 terminals from the M6 layer of the medulla (*Figure 1A–C*); 30–60% of R7 axons were mistargeted to the M3 layer (*Figure 1D–F*). This phenotype appears to arise during the second stage of R7 targeting, when filopodia are stabilized to form synapses (*Ting et al., 2005*; *Özel et al., 2019*). R7 axons targeted correctly to their temporary layer at 40 hr after puparium formation (APF) when *CG6024* was knocked down, but many terminals did not reach or were not stabilized in their permanent target layer, M6, at 60 hr APF (*Figure 1G–K*). We named the gene *lost and found* (*loaf*) based on the failure of R7 axons lacking *loaf* to find the right target layer and on the rescue of this phenotype discussed below. The Loaf protein contains extracellular CUB and LDLa domains and a predicted transmembrane domain (*Figure 2A*), making it a candidate to directly mediate target recognition by R7.

### *loaf* mutant R7 axons show targeting defects only when *loaf* is present in other cells

In the experiments above, we used two independently generated RNAi lines targeting the same region of the gene to knock down *loaf* (*Figure 2A*), both of which produced similar R7 mistargeting phenotypes (*Figure 1D*). To confirm that this phenotype was due to loss of *loaf* rather than an off-target effect of the RNAi, we used the CRISPR-Cas9 system to generate deletion alleles that removed the LDLa, transmembrane and cytoplasmic domains of the protein (*Figure 2A,I*). The sgRNAs were directed against a region of the gene distinct from the RNAi target sequence, and using them to delete the *loaf* gene in the eye by somatic CRISPR reproduced the R7 targeting defect (*Figure 2A,B,H*). Surprisingly, germline removal of *loaf* resulted in homozygous mutant flies that were viable and showed largely normal R7 targeting (*Figure 2C,D,H*), indicating that global loss of *loaf* does not affect this process. Expressing *loaf* RNAi had no effect in this *loaf* mutant background (*Figure 2E,H*), confirming that the RNAi phenotype was due to its effect on *loaf* rather than another gene. Together, these results indicate that the phenotype caused by removing *loaf* from the eye is dependent on the presence of *loaf* in the optic lobes. R7 targeting may therefore depend on the amount of Loaf in R7 relative to other cells rather than its absolute presence or absence.

To test this hypothesis, we generated clones of cells in the eye that were homozygous for *loaf* deletion alleles in an otherwise heterozygous background. As predicted, these showed mistargeting of R7 axons to the M3 layer (*Figure 2F,H*, *Figure 2—figure supplement 1A*). The mistargeting was significantly rescued by expressing either HA-tagged or untagged Loaf within the mutant clones (*Figure 2G,H*, *Figure 2—figure supplement 1B–D*), confirming that it is due to loss of *loaf* from photoreceptors. These results could be explained if correct targeting depends on the relative levels of Loaf in R7 and another cell type. Loss of Loaf in R7 when it is present in the other cell type would cause mistargeting. When Loaf is absent from all cells, redundant mechanisms may be sufficient to maintain R7 terminals in the correct layer.

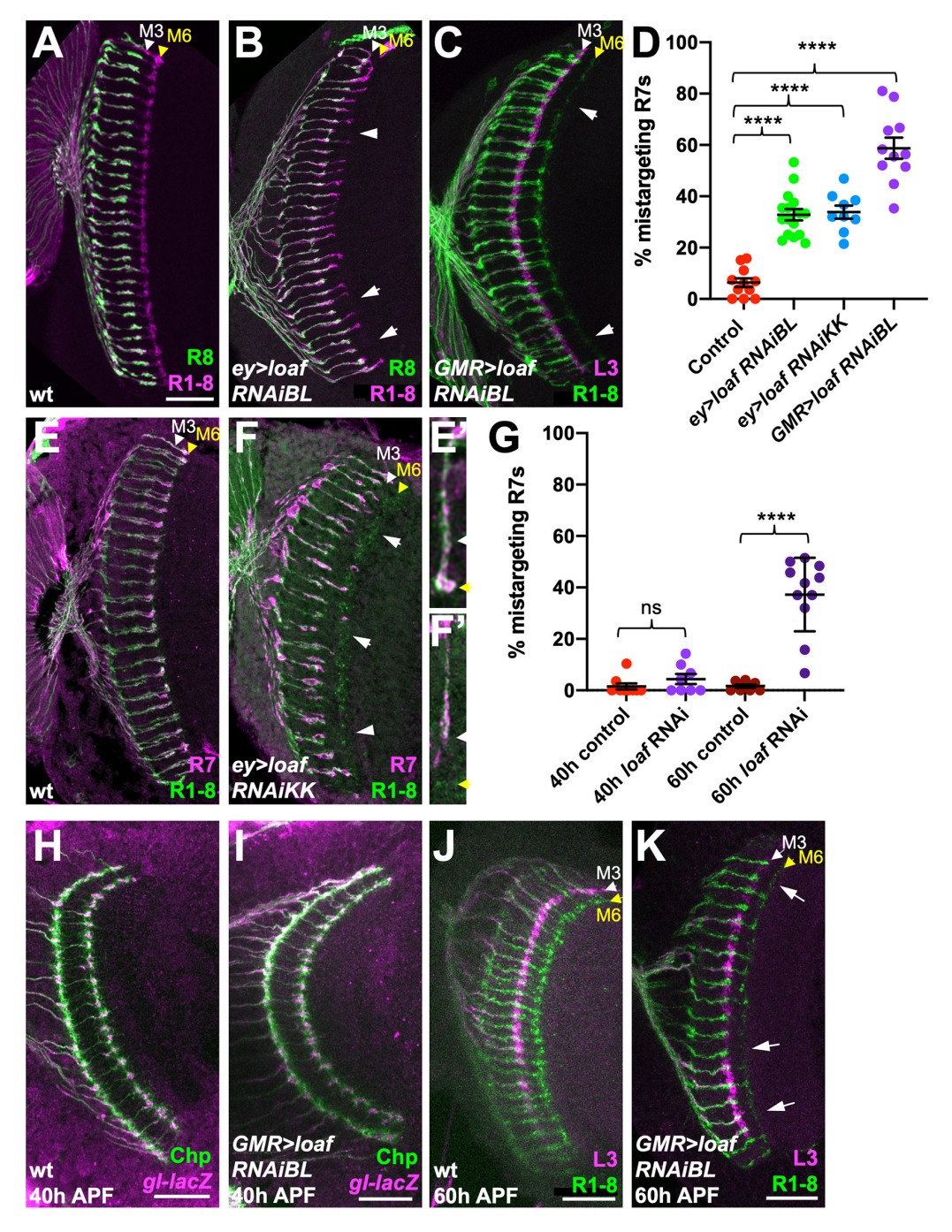

Figure 1. *loaf* RNAi in photoreceptors causes R7 mistargeting. (A–C, E, F) cryostat sections of adult heads stained for Chaoptin (Chp) to label all photoreceptor axons (magenta in A, B, green in C, E, F), *Rh5-GFP* and *Rh6-GFP* to label R8 (green in A, B), *22E09-LexA* driving *LexAop-myr-tdTomato* to label lamina neuron L3, which projects to the M3 layer (magenta in C) or *panR7-lacZ* to label R7 (magenta in E, F). (A, E) wild type; (B) *ey3.5-FLP, Act>CD2>GAL4; UAS-dcr2; UAS-loaf RNAiBL (P{TRiP.JF03040}attP2)*; (C) *GMR-GAL4, UAS-dcr2; UAS-loaf RNAiBL*; (F) *ey3.5-FLP, Act>CD2>GAL4; UAS-dcr2; UAS-loaf RNAiKK (P{KK112220}VIE-260B)*. Arrows show examples of R7 mistargeting. White arrowheads indicate the M3 layer and yellow arrowheads the M6 layer. (E', F') show enlargements of single R7 axons from (E, F). (D) Quantification of the percentage of R7 axons that failed to reach the M6 layer in the same genotypes. n = 11 (control, *GMR > RNAiBL*), 16 (*ey>RNAiBL*), or 9 (*ey>RNAiKK*). ****p<0.0001 by unpaired t-test. Error bars show mean ± standard error of the mean (SEM) in this and all other graphs. (H–K) Pupal brains stained for Chp (green) and *glass* (*gl*)-*lacZ*, which labels all photoreceptor axons (magenta in H, I) or *22E09-LexA* driving *LexAop-myr-tdTomato* to label L3 neuronal processes in the M3 layer (magenta in J, K). (H, I) Forty hr after puparium formation (APF); (J, K) 60 hr APF. (H, J) wild type (*GMR-GAL4, UAS-dcr2/+*); (I, K) *GMR-GAL4, UAS-dcr2; UAS-loaf RNAiBL*. Loss of *loaf* does not prevent the initial targeting of R7 axons to their temporary layer at 40 hr APF, but many axons fail to project beyond that layer at

*Figure 1 continued on next page*

*Figure 1 continued*

60 hr APF. (**G**) Quantification of the percentage of R7 axons that did not reach the appropriate layer for these genotypes and stages. n = 9 (40 h control), 8 (40 h RNAi, 60 hr control), or 11 (60 h RNAi). ****, p<0.0001 by unpaired t-test with Welch's correction; ns, not significant. Scale bars, 20 μm.
The online version of this article includes the following source data for figure 1:

**Source data 1.** Data shown in *Figure 1D and G*.

## Loaf levels in Dm8, the major synaptic target of R7, do not affect R7 targeting

The medulla interneuron Dm8, which mediates the preference for ultraviolet over visible light, was reported to be the major postsynaptic target of R7 (*Gao et al., 2008*; *Takemura et al., 2013*; *Ting et al., 2014*). We therefore considered the hypothesis that R7 and its postsynaptic partner Dm8 must both express Loaf to form a stable connection (*Figure 3F*). We first determined the effect of removing *loaf* function from Dm8. Expressing *loaf* RNAi or Cas9 and *loaf* sgRNAs in neurons that include Dm8 cells with *DIP-γ-GAL4* or *traffic jam* (*tj*)-GAL4 (*Carrillo et al., 2015*; *Courgeon and Desplan, 2019*) did not cause any R7 targeting phenotype (*Figure 3—figure supplement 1A,B*). As it was difficult to assess the reduction in Loaf levels caused by these manipulations, we generated *loaf* mutant clones in the brain and labeled the mutant Dm8 cells with *ortc2b-GAL4* (*Ting et al., 2014*). R7 axons that contacted the dendrites of mutant Dm8 cells correctly reached the M6 layer, and there was no obvious defect in the position or morphology of the mutant Dm8 dendrites (*Figure 3A,B*).

We predicted that expressing Loaf in Dm8 cells in a *loaf* mutant background would result in a mismatch between R7 and Dm8 that would be similar to removing *loaf* from R7 in a wild-type background (*Figure 3F*). We tested this by expressing UAS-LoafHA in *loaf* mutant flies with the Dm8 drivers *DIP-γ-GAL4*, *tj-GAL4* and *drifter* (*drf*)-GAL4 (*Hasegawa et al., 2011*; *Carrillo et al., 2015*; *Courgeon and Desplan, 2019*), as well as a combination of *tj-GAL4* and *DIP-γ-GAL4*. However, we did not observe significant levels of R7 mistargeting (*Figure 3C–E*, *Figure 3—figure supplement 1C,D*), arguing against a requirement for matching Loaf levels in R7 and Dm8.

## Loaf levels in cholinergic and glutamatergic neurons influence R7 targeting

Since the presence or absence of Loaf in Dm8 did not appear to affect R7 targeting, we searched for other Loaf-expressing cells that might interact with R7. We used several methods to examine the location of Loaf expression in the brain. RNA-Seq analysis of sorted cell types in the adult brain revealed widespread expression of *loaf*, although at varying levels (*Konstantinides et al., 2018*; *Davis et al., 2020*). However, Loaf translation in photoreceptors reaches its maximum at mid-pupal stages, when R7 axons are targeting the M6 layer (*Ting et al., 2005*; *Zhang et al., 2016*), so adult expression levels in other cells might not be reflective of this developmental stage. At pupal stages, we observed that a GFP protein trap insertion in *loaf* was expressed in many cells in the medulla (*Figure 2A*, *Figure 4—figure supplement 1A*). Finally, we generated an antibody that recognizes the cytoplasmic domain of Loaf (*Figure 2I*). As this antibody cross-reacted with another protein present in the cell bodies of medulla neurons (*Figure 4C*), we could only evaluate Loaf expression within the neuropil. In pupal brains, Loaf was enriched in specific layers of the medulla neuropil and also in R7 axons and terminals (*Figure 4A*; *Figure 4—figure supplement 1C*). This staining was absent in *loaf* mutants (*Figure 4C*), and the enrichment in R7 processes was specifically lost when *loaf* RNAi was expressed with *GMR-GAL4* (*Figure 4B*; *Figure 4—figure supplement 1D*). The Loaf protein trap was primarily present in cell bodies and was not visibly enriched in R7 axons; we believe that this insertion disrupts the normal localization of the protein, as clones homozygous for the insertion showed R7 mistargeting (*Figure 4—figure supplement 1A,B*). When misexpressed in photoreceptors, LoafHA was efficiently transported to R7 axons and terminals (*Figure 4—figure supplement 1F*). Consistent with our findings, recent single-cell RNA-Seq data from dissociated optic lobes show that significant *loaf* expression is present in almost every cluster throughout the pupal stage, although its levels are generally lower in clusters identified as glia. *loaf* expression in photoreceptors is highest at P40 and P50, but declines at later stages (*Kurmangaliyev et al., 2020*; *Özel et al., 2021*).

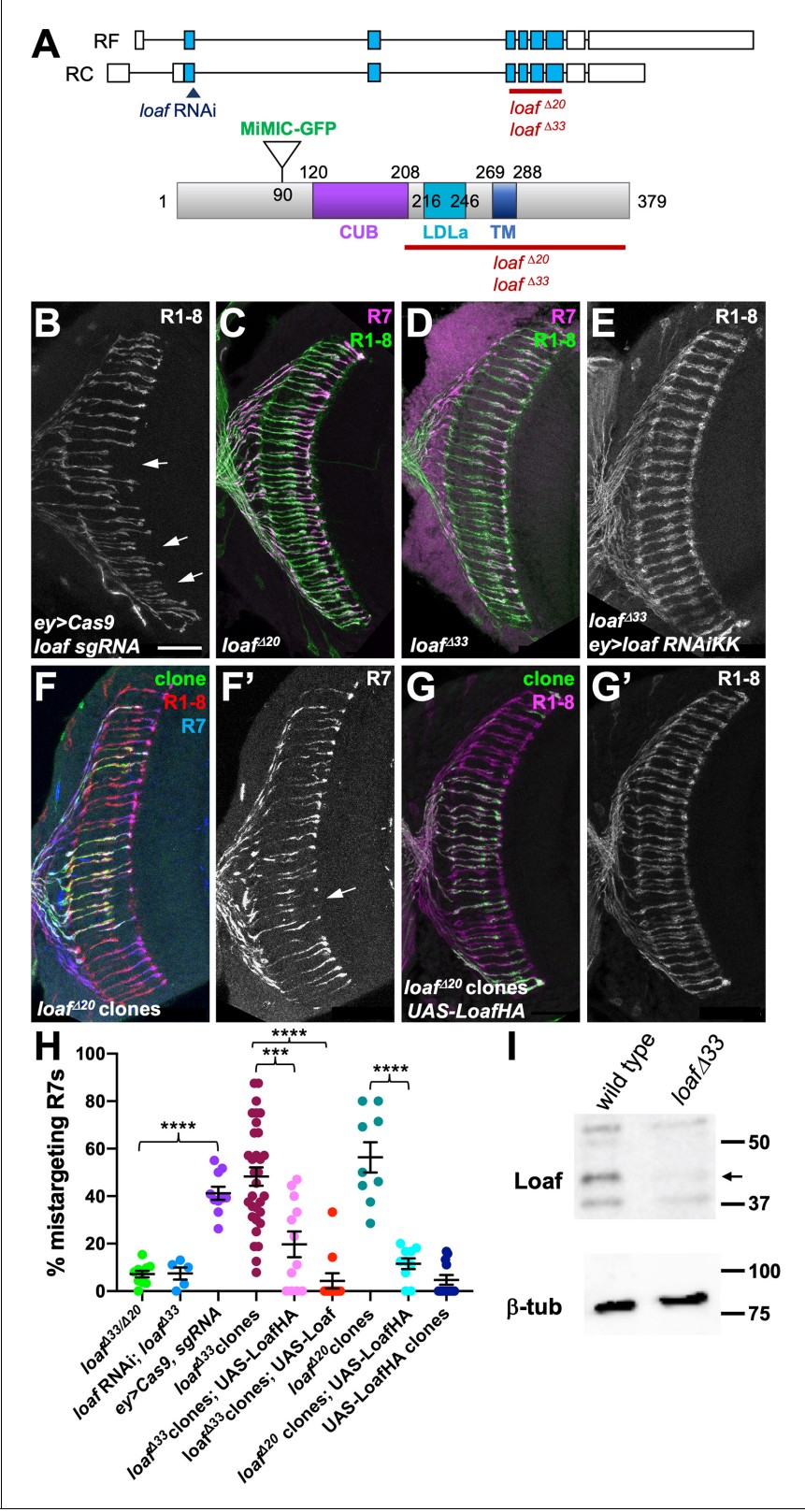

**Figure 2.** R7 is only affected by eye-specific loss of *loaf*. (**A**) Diagrams of the *loaf* gene and protein. Coding exons, which are identical for the two isoforms, are shown as blue boxes and non-coding exons as white boxes. The region targeted by both RNAi lines, the MiMIC GFP insertion and the extent of the *loaf^Δ20* and *loaf^Δ33* deletions are indicated. These two deletions were independently generated and have minor sequence differences around the cut site. TM, transmembrane domain. (**B–G**) cryostat sections of adult heads stained for Chp (B, E, G', green in C, D, red in F, magenta in *Figure 2 continued on next page*

*Figure 2 continued*

G), *panR7-lacZ* (F', magenta in C, D, blue in F), and GFP (green in F, G). (B) *ey3.5-FLP, Act>CD2>GAL4; loaf sgRNAs; UAS-Cas9P2*; (C) *loaf^Δ20^* homozygote; (D) *loaf^Δ33^* homozygote; (E) *ey3.5-FLP, Act>CD2>GAL4; UAS-dcr2/UAS-loaf RNAiKK; loaf^Δ33^*; (F) *loaf^Δ20^* clones positively labeled with *lGMR-GAL4, UAS-GFP*; (G) *loaf^Δ20^* clones expressing *UAS-LoafHA* with *lGMR-GAL4*, positively labeled with GFP. Scale bar, 20 μm. (H) quantification of the percentage of R7 axons that failed to reach the M6 layer in the indicated genotypes. n = 10 (*loaf^Δ33^/loaf^Δ20^; ey>Cas9, sgRNA; loaf^Δ20^* clones, *UAS-LoafHA*), 5 (*loafRNAi; loaf^Δ33^*), 32 (*loaf^Δ33^* clones), 12 (*loaf^Δ33^* clones, *UAS-LoafHA*; wild type clones, *UAS-LoafHA*), 11 (*loaf^Δ33^* clones, *UAS-Loaf*), or 9 (*loaf^Δ20^* clones). Error bars show mean ± SEM. ***, p<0.0005; ****, p<0.0001 by unpaired t-test, with Welch's correction when variances are significantly different. *loaf* homozygotes show little R7 mistargeting, but are resistant to the effect of *loaf* RNAi. R7 mistargeting is observed when *loaf* sgRNAs and Cas9 are expressed in the eye, and in clones homozygous for *loaf* alleles. This clonal phenotype is rescued by expressing *UAS-LoafHA* or *UAS-Loaf* in the mutant cells. (I) Western blot of extracts from wild type and *loaf^Δ33^* larval brains using an antibody to the cytoplasmic domain of Loaf and β-tubulin antibody as a loading control. Loaf protein (arrow) is absent in *loaf^Δ33^* mutants.

The online version of this article includes the following source data and figure supplement(s) for figure 2:

**Source data 1.** Data shown in *Figure 2H*.
**Figure supplement 1.** R7 mistargeting in *loaf* mutant clones is rescued by tagged or untagged Loaf.

Because these data did not identify a specific cell type that would be most likely to interact with R7 using Loaf, we tested whether R7 mistargeting could be induced by expressing Loaf in broad categories of cells in a *loaf* mutant background. We observed no phenotype when Loaf was expressed in glia with *repo-GAL4*, or in neuronal populations that expressed *homothorax* (*hth*)-*GAL4*, *brain-specific homeobox* (*bsh*)-*GAL4*, or *Visual system homeobox* (*Vsx*)-*GAL4* (*Hasegawa et al., 2011*; *Li et al., 2013*; *Erclik et al., 2017*; *Figure 4D*). Expressing Loaf in photoreceptors with *ey3.5-FLP, Act>CD2>GAL4* in a *loaf* mutant background likewise had no effect on R7 (*Figure 4D*), indicating that the presence of Loaf in R7 when it is absent in other cells did not impede its targeting. However, we did observe a significant level of R7 mistargeting when Loaf was expressed in neurons that expressed *apterous* (*ap*)-*GAL4*, which is active from the third larval instar (*Morante et al., 2011*; *Li et al., 2013*) or in cholinergic neurons with *Choline acetyltransferase* (*ChAT*)-*GAL4*, which is active from mid-pupal stages (*Meissner et al., 2019*), in a *loaf* mutant background (*Figure 4D–F*). *ap* is expressed in the majority of cholinergic neurons in the medulla (*Konstantinides et al., 2018*), supporting the idea that cells in this population use Loaf to interact with R7. R7 targeting defects also occurred when Loaf was expressed in glutamatergic neurons in a *loaf* mutant background with *Vesicular glutamate transporter* (*VGlut*)-*GAL4*, which is active from early pupal stages (*Meissner et al., 2019*; *Figure 4D*; *Figure 4—figure supplement 1E*), indicating that more than one type of neuron interacts with R7 through Loaf.

## Loaf levels in the synaptic partners Tm5a/b and Dm9 influence R7 targeting

The populations of cholinergic and glutamatergic neurons include the major synaptic targets of R7, suggesting the possibility that Loaf acts in these cells to influence their interactions with R7. The synapses that R7 forms with Dm8 also include the cholinergic output neurons Tm5a and Tm5b (*Gao et al., 2008*; *Karuppudurai et al., 2014*; *Menon et al., 2019*; *Davis et al., 2020*). To test the importance of Tm5a/b neurons we used *GMR9D03-GAL4*, which is expressed in a subset of these cells from early in development (*Han et al., 2011*; *Figure 5—figure supplement 1A,B*) to express Loaf in a *loaf* mutant background. This again produced significant R7 mistargeting (*Figure 5A,H*), consistent with the hypothesis that Loaf levels in Tm5a/b influence R7. Although *GMR9D03-GAL4* is also expressed in lamina neurons L2 and L3 (*Akin et al., 2019*), restoring Loaf only in lamina neurons with *GH146-GAL4* (*Schwabe et al., 2014*) did not affect R7 (*Figure 5H*). Importantly, *loaf* mutant Tm5a/b cells did not have obvious morphological defects or cause R7 mistargeting (*Figure 5D,E*).

Among glutamatergic neurons, both Dm8 and Dm9 are synaptic partners of R7. Dm9 is a multicolumnar neuron that tracks R7 axons closely and mediates inhibition between neighboring ommatidia (*Nern et al., 2015*; *Heath et al., 2020*). The transcription factors Vestigial (Vg) and Defective proventriculus (Dve) are strongly enriched in Dm9 cells (*Davis et al., 2020*), and *dve-GAL4* drives expression in Dm9 (*Figure 5—figure supplement 1C*), although *vg-GAL4* expression was not detectable in the adult brain. When used to express Loaf in a *loaf* mutant background, neither driver alone significantly affected R7 targeting, but the combination had a significant effect (*Figure 5B,H*), making Dm9 a candidate to provide Loaf that affects R7 targeting. Again, *loaf* mutant Dm9 cells and

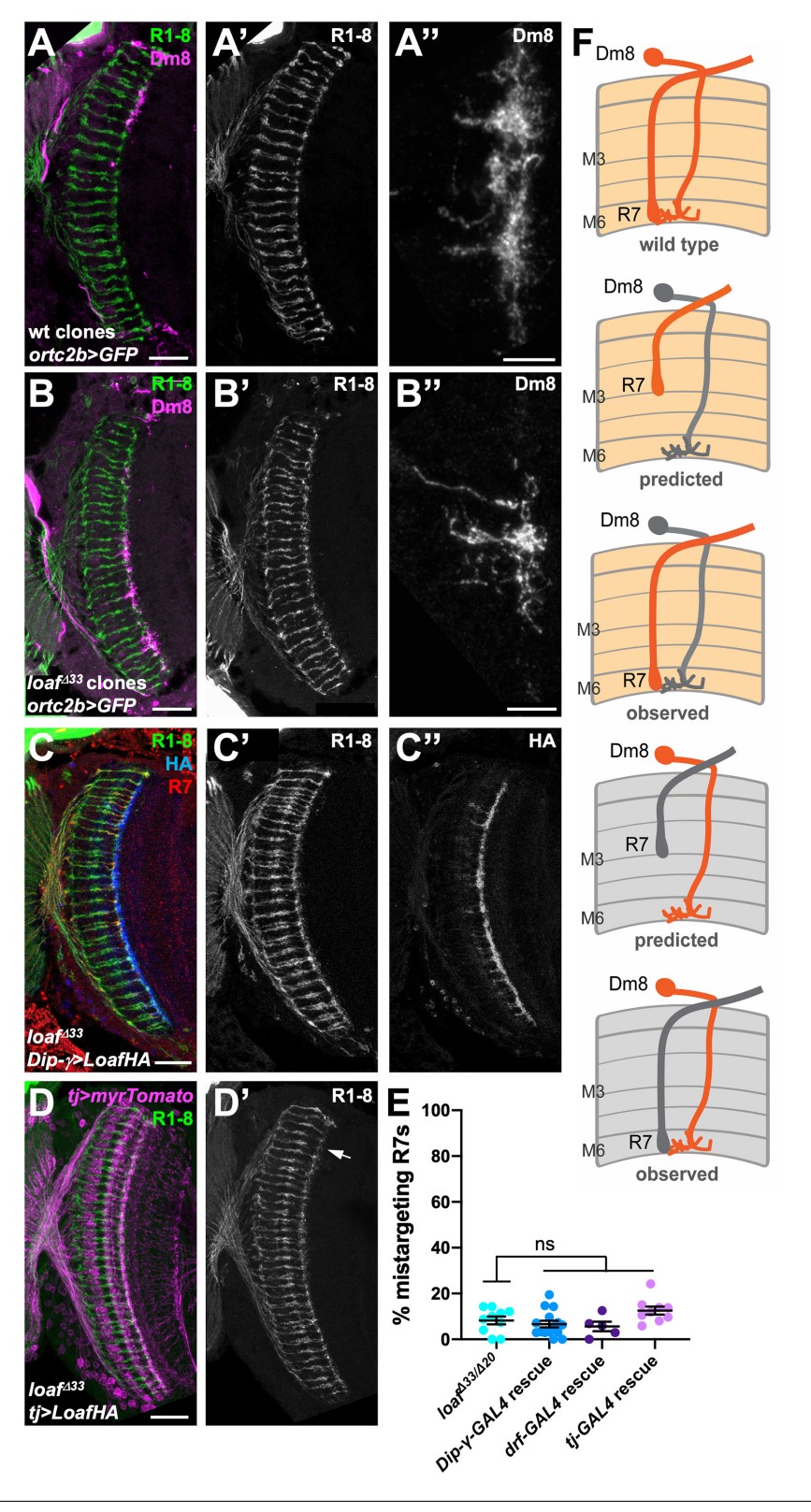

**Figure 3.** Changing the level of Loaf in Dm8 does not affect R7 targeting. (A–D) cryostat sections of adult heads stained for Chp (A'-D', green in A-D), GFP (A'', B'', magenta in A, B), *panR7-lacZ* (red in C), HA (C'', blue in C), or myrTomato (magenta in D). (A) wild type clones in which Dm8 is labeled with *ortc2b-GAL4, UAS-CD8GFP*; (B) *loaf^Δ33* clones in which Dm8 is labeled with *ortc2b-GAL4, UAS-CD8GFP*. A'' and B'' show enlargements of labeled Dm8 dendrites. *loaf* mutant Dm8 dendrites and the R7 axons that target them have the normal position and morphology. (C) *UAS-LoafHA; DIP-γ-GAL4,*

*Figure 3 continued on next page*

Figure 3 continued

*loaf*$^{\Delta33}$/*loaf*$^{\Delta33}$; (D) *tj-GAL4/UAS-LoafHA; loaf*$^{\Delta33}$/*loaf*$^{\Delta33}$. The arrow in (D') indicates minor R7 mistargeting that was not statistically significant. Scale bars, 20 µm (A–D), 5 µm (A'', B''). (E) quantification of the percentage of R7 axons that failed to reach the M6 layer in the indicated genotypes. n = 10 (*loaf*$^{\Delta33}$/*loaf*$^{\Delta20}$; *DIP-γ -GAL4* rescue; *tj-GAL4* rescue), or 5 (*drf-GAL4* rescue). Error bars show mean ± SEM. ns, not significant by unpaired t-test. Expressing Loaf in Dm8 neurons in a *loaf* mutant does not cause R7 mistargeting. (F) diagrams explaining the predicted results if Loaf expression in R7 has to match its expression in Dm8. R7 and Dm8 both express Loaf (orange), which is also present in other cells in the brain. Removing *loaf* from Dm8 (gray) or expressing Loaf in Dm8 in a *loaf* mutant (gray in R7 and brain) would cause a mismatch and is predicted to result in R7 mistargeting. However, (A–E) show that there is no mistargeting in these situations (observed), indicating that Loaf does not act in Dm8 to regulate R7 targeting.

The online version of this article includes the following source data and figure supplement(s) for figure 3:

**Source data 1.** Data shown in *Figure 3E*.
**Figure supplement 1.** Changing Loaf levels in Dm8 has no effect.

their presynaptic R7 axons appeared normal (*Figure 5F,G*). Finally, we tested whether a contribution of Dm8 might be detectable in combination with other R7 synaptic target cells by restoring Loaf to *loaf* mutants with both *ap-GAL4* and *tj-GAL4*. This produced significantly more R7 mistargeting than *ap-GAL4* alone (*Figure 5C,H*), suggesting that Dm8 or another *tj-GAL4* expressing neuron such as Dm11 (*Courgeon and Desplan, 2019*), which also projects to the M6 layer (*Nern et al., 2015*; *Figure 5—figure supplement 1D*), may contribute to the pool of Loaf that influences R7 targeting. However, Tm5a/b and Dm9 appear to play a more significant role (*Figure 5I*). Overexpressing Loaf with the *GMR9D03-GAL4*, *dve-GAL4* and *vg-GAL4*, or *ap-GAL4* and *tj-GAL4* drivers in a wild-type background did not cause R7 mistargeting (*Figure 5—figure supplement 2*); because Loaf is normally enriched in R7 terminals, it is possible that the Loaf levels produced in the processes of synaptic partner cells in these overexpression experiments did not exceed those present in R7.

## Loaf may act indirectly through cell-surface molecules

To determine whether Loaf could function as a homophilic cell adhesion molecule, we transfected HA-tagged Loaf into S2 cells and conducted cell aggregation assays (*Ting et al., 2005*; *Astigarraga et al., 2018*). We did not observe significant aggregation of the transfected cells, although the positive control Sidekick (Sdk) (*Astigarraga et al., 2018*) induced aggregation under the same conditions (*Figure 6A,B*, *Figure 6—figure supplement 1A*). Unlike Sdk, neither tagged nor untagged Loaf showed strong localization to the plasma membrane; most Loaf was present in punctate structures inside the cells (*Figure 6C–E*). These structures showed partial colocalization with Hepatocyte growth-factor-regulated tyrosine kinase substrate (Hrs) and Rab7 (*Figure 6D,E*), two markers of late endosomes, but did not colocalize with the recycling endosome marker Rab11-GFP (*Figure 6—figure supplement 1C*). When expressed in the retina in vivo, LoafHA also partially colocalized with Rab7 and Hrs, but not with the lysosomal markers ADP-ribosylation factor-like 8 (Arl8) or Vacuolar H$^+$-ATPase 55kD subunit (Vha55) (*Figure 6F,G,J*), and untagged Loaf again showed a similar localization (*Figure 6—figure supplement 1D*). In clones of cells mutant for the ESCRT complex component *Tumor susceptibility gene 101* (*TSG101*), endocytosed proteins such as Notch accumulate in late endosomes (*Moberg et al., 2005*), and we found that LoafHA colocalized with Notch (*Figure 6—figure supplement 1E*), confirming its presence in the endocytic pathway.

GFP-tagged endogenous Loaf appeared to localize to the cytoplasm of all photoreceptors, but unlike overexpressed Loaf, it was primarily found close to the plasma membrane rather than in late endosomes (*Figure 6—figure supplement 1B*); as noted above, this tag disrupts the function of the Loaf protein. As a more stringent test of whether Loaf ever reaches the plasma membrane, we transfected S2 cells with a form of Loaf tagged at its extracellular N-terminus with the V5 epitope, and incubated live cells with antibodies to V5. No staining was observed in these conditions (*Figure 6H*). As controls for this experiment, V5 staining was detected in cells that were fixed and permeabilized, and antibodies to HA detected cotransfected HASdk on the surface of live cells as well as in vesicles internalized during the incubation (*Figure 6H,I*). These results suggest that Loaf is not itself a cell surface adhesion molecule, but could regulate the trafficking or cell surface localization of proteins involved in cell adhesion or synapse formation.

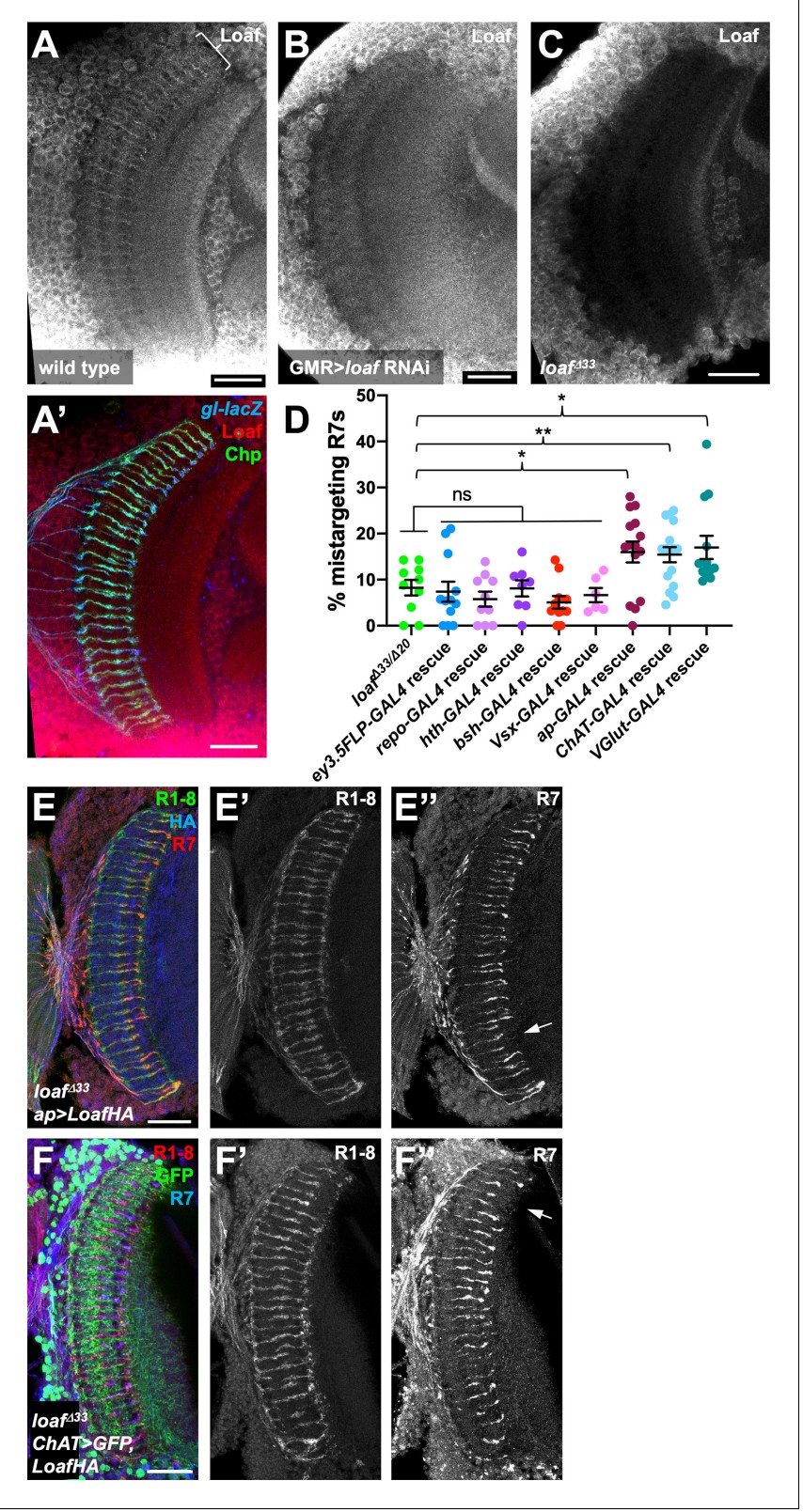

**Figure 4.** Loaf levels in cholinergic and glutamatergic neurons influence R7 targeting. (**A–C**) Pupal brains at 60 hr APF stained for Loaf (A-C, red in A'), Chp (green in A') and *gl-lacZ* (blue in A'). (**A**) wild type; (**B**) *GMR-GAL4, UAS-dcr2; UAS-loaf RNAiBL*; (**C**) *loaf^Δ33^*. Loaf antibody staining in the medulla neuropil is absent in the *loaf* mutant (**C**). Enriched staining in R7 axons (bracket in A) is lost when *loaf* is knocked down in photoreceptors (**B**). The antibody appears to cross-react with a protein present in medulla cell bodies, as this staining is still present in *loaf* mutant brains (**C**). (**D**) quantification

*Figure 4 continued on next page*

Figure 4 continued

of the percentage of R7 axons that failed to reach the M6 layer in the indicated genotypes. n = 10 (loaf$^{\Delta33}$/loaf$^{\Delta20}$; repo-GAL4 rescue), 12 (ey3.5-FLP, Act>CD2>GAL4 rescue), 8 (hth-GAL4 rescue), 11 (bsh-GAL4 rescue), 6 (Vsx-GAL4 rescue), 15 (ap-GAL4 rescue), 16 (ChAT-GAL4 rescue), or 13 (vGlut-GAL4 rescue). Error bars show mean ± SEM. *, p<0.05; **, p<0.01; ns, not significant by unpaired t-test. Expressing Loaf in cholinergic or glutamatergic neurons or in the precursors of cholinergic neurons with ap-GAL4 in a loaf mutant causes R7 mistargeting. (E, F) cryostat sections of adult heads stained for Chp (E', F', green in E, red in F), panR7-lacZ (E'', F'', red in E, blue in F), HA (blue in E), or GFP (green in F). (E) ap-GAL4/UAS-LoafHA; loaf$^{\Delta33}$; (F) ChAT-GAL4, UAS-CD8GFP/UAS-LoafHA; loaf$^{\Delta33}$. Scale bars, 20 µm.

The online version of this article includes the following source data and figure supplement(s) for figure 4:

**Source data 1.** Data shown in *Figure 4D*.
**Figure supplement 1.** Loaf is expressed in many cells and enriched in R7 terminals.

## Loaf interacts with Lar and enhances the function of Lrp4

We next searched for candidate proteins that might be regulated by Loaf. One possibility we considered was Lar, an RPTP that is required for normal R7 targeting (*Clandinin et al., 2001*; *Maurel-Zaffran et al., 2001*; *Hofmeyer and Treisman, 2009*). *Lar* acts in R7 and not the target region (*Clandinin et al., 2001*; *Maurel-Zaffran et al., 2001*), so its ligand, which remains unknown, would also have to be regulated by Loaf to account for the effect of Loaf in synaptic partners of R7. To test for a genetic interaction between *loaf* and *Lar*, we knocked down these genes using the photoreceptor driver *long GMR-GAL4* (*lGMR-GAL4*) (*Wernet et al., 2003*). Expression of either *loaf* RNAi or *Lar* RNAi with this driver affected only a subset of R7 axons, but simultaneous expression of both RNAi lines had a synergistic effect, causing almost all R7 axons to terminate in the M3 layer (*Figure 7A–D*). This suggests that Loaf and Lar are involved in the same process. Similarly, *loaf* knockdown enhanced the mistargeting phenotype of mutations in the downstream gene *Liprin-α*, although this effect could simply be additive (*Figure 7—figure supplement 1G–J*). Overexpression of Lar in photoreceptors, either alone or together with Loaf, did not cause any significant defects (*Figure 7—figure supplement 1A,B,D*). However, overexpression of Lar in *loaf* mutant photoreceptors could rescue R7 targeting (*Figure 7—figure supplement 1C,D*), indicating that Lar can compensate for the lack of *loaf* and is thus unlikely to be its primary effector. Consistent with this conclusion, we found that *loaf* was not required for HA-tagged Lar to be transported into photoreceptor axons (*Figure 7—figure supplement 1E,F*).

We also investigated LDL receptor related protein 4 (Lrp4), based on its role as a presynaptic organizer in the olfactory system (*Mosca et al., 2017*), its postsynaptic signaling function at the vertebrate neuromuscular junction (*Yumoto et al., 2012*), and the requirement for chaperones to promote the trafficking of other LDL family members (*Culi et al., 2004*; *Wagner et al., 2013*). We found evidence that the level of Lrp4 can affect R7 targeting and that its effect on R7 is regulated by Loaf. Overexpressing Lrp4 in photoreceptors caused R7 axons to contact each other or hyperfasciculate either in the M3 or M6 layers of the medulla (*Figure 7E,H*). These defects were more severe, and included overshooting of the M6 layer by some R7 axons, when Lrp4 was coexpressed with Loaf (*Figure 7F,H*), but were almost absent when Lrp4 was expressed in *loaf* mutant photoreceptors (*Figure 7G,H*). Lrp4 overexpression also resulted in abnormal numbers and arrangements of cone and pigment cells in the retina (*Figure 7—figure supplement 2A*). Again, these defects were more severe when Lrp4 was coexpressed with Loaf, and were not observed when Lrp4 was expressed in *loaf* mutant cells (*Figure 7—figure supplement 2B,C*). Although Lrp4HA had a more granular appearance in *loaf* mutant than in wild-type photoreceptor cell bodies (*Figure 7—figure supplement 2D,F*), it was still transported into their axons (*Figure 7—figure supplement 2E,G*), and its level of expression appeared unaffected (*Figure 7—figure supplement 2H*).

Despite the effect of Loaf on Lrp4 function, Lrp4 is unlikely to fully explain the effects of *loaf* on R7 targeting, as R7 axons projected normally in *Lrp4* mutant clones (*Figure 7—figure supplement 2I*). Moreover, expressing *loaf* RNAi in photoreceptors resulted in R7 mistargeting even in an *Lrp4* null mutant background (*Figure 7—figure supplement 2J,K*). These results show that Loaf can affect the function of cell surface proteins, and suggest that it could act by regulating Lrp4 and/or other cell surface molecules that act as a readout of its levels to control the interactions between R7 and its postsynaptic partners.

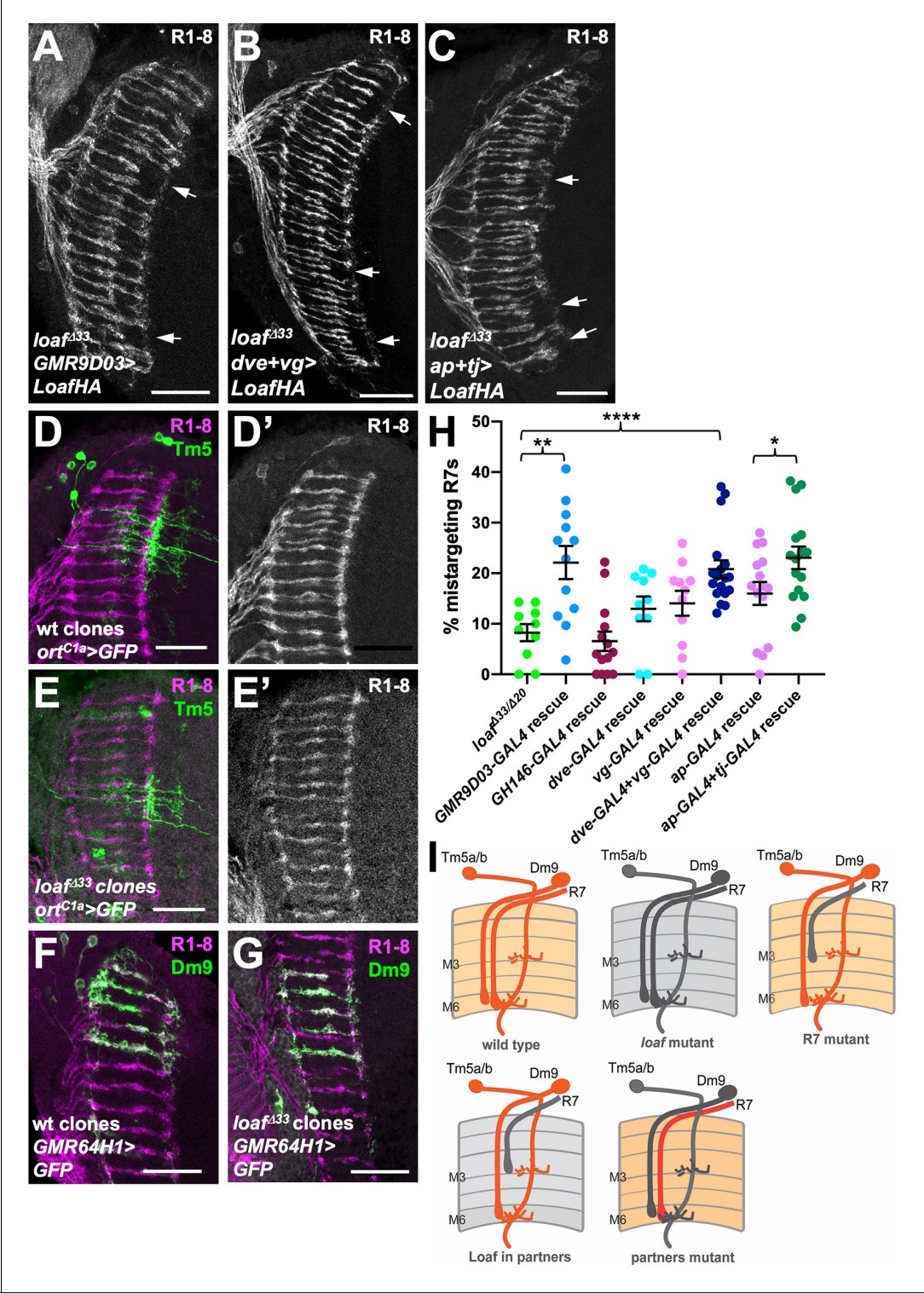

**Figure 5.** Expressing Loaf in synaptic partners of R7 in a *loaf* mutant causes mistargeting. (**A–C**) cryostat sections of adult heads stained for Chp. (**A**) *UAS-LoafHA; GMR9D03-GAL4, loaf$^{Δ33}$/loaf$^{Δ33}$*; (**B**) *dve-GAL4, vg-GAL4/UAS-LoafHA; loaf$^{Δ33}$*; (**C**) *ap-GAL4, tj-GAL4/UAS-LoafHA; loaf$^{Δ33}$*. Expressing Loaf in populations of neurons that form synapses with R7 in a *loaf* mutant background causes R7 mistargeting. (**D–G**) cryostat sections of adult heads in which clones generated with *hs-FLP* are labeled in green with UAS-CD8-GFP and R1-8 are stained with anti-Chp (D', E', magenta in D-G). (**D**) wild type and (**E**) *loaf$^{Δ33}$* mutant clones in which Tm5a/b/c and Tm20 are labeled with *ort$^{C1a}$-GAL4*. The genotypes are (**D**) *hsFLP, UAS-GFP; ortc1a-GAL4/CyO; FRT80/tub-GAL80, FRT80*; (**E**) *hsFLP, UAS-GFP; ortc1a-GAL4/CyO; loaf$^{Δ33}$, FRT80/tub-GAL80, FRT80*. (**F**) wild type and (**G**) *loaf$^{Δ33}$* mutant clones in which Dm9 cells are labeled with *GMR64H1-GAL4*. The genotypes are (**F**) *hsFLP, UAS-GFP; GMR64H1-GAL4, FRT80/tub-GAL80, FRT 80*; (**G**) *hsFLP, UAS-GFP;*

*Figure 5 continued*

*GMR64H1-GAL4, loaf$^{\Delta33}$, FRT80/tub-GAL80, FRT 80*. The morphologies of wild type and *loaf* mutant Tm5 and Dm9 cells appear similar. (**H**) Quantification of the percentage of R7 axons that failed to reach the M6 layer in the indicated genotypes. n = 10 (*loaf$^{\Delta33}$/loaf$^{\Delta20}$; dve-GAL4* rescue), 12 (*GMR9D03-GAL4* rescue), 14 (*GH146-GAL4* rescue), 11 (*vg-GAL4* rescue), 18 (*dve-GAL4 + vg* GAL4 rescue), 15 (*ap-GAL4* rescue), or 16 (*ap-GAL4 +tj* GAL4 rescue). Error bars show mean ± SEM. *p<0.05; **p<0.01; ****p<0.0001 by unpaired t-test. (**I**) model showing that the presence of Loaf in Dm9 or Tm5a/b when *loaf* is absent in R7 causes R7 mistargeting. Scale bars, 20 μm.

The online version of this article includes the following source data and figure supplement(s) for figure 5:

**Source data 1.** Data shown in *Figure 5H*.
**Figure supplement 1.** GAL4 drivers for Tm5a/b and Dm9.
**Figure supplement 2.** Overexpression of Loaf in the synaptic partners of R7 does not cause mistargeting.
**Figure supplement 2—source data 1.** Data shown in *Figure 5—figure supplement 2D*.

## Discussion

### Tm5a/b and Dm9 cells provide cues for R7 targeting

The layered arrangement of neuronal processes in the medulla makes R7 axon targeting a sensitive model system in which to elucidate how growth cones select the correct postsynaptic partners. However, it has not been clear which cells are responsible for retaining R7 axons in the M6 layer. The RPTP Lar, which forms a hub for the assembly of presynaptic structures through the adaptor protein Liprin-α (*Choe et al., 2006*; *Hofmeyer et al., 2006*; *Takahashi and Craig, 2013*; *Bomkamp et al., 2019*), acts in R7 to stabilize filopodia in the M6 layer by promoting synapse formation (*Clandinin et al., 2001*; *Maurel-Zaffran et al., 2001*; *Özel et al., 2019*). Our findings that *Lar* and *loaf* show a strong genetic interaction and that *Lar* overexpression can rescue the loss of *loaf* suggest that like Lar, Loaf stabilizes synaptic contacts. Although Lar family RPTPs can recognize a variety of ligands (*Han et al., 2016*), the ligand involved in R7 targeting and its cellular source remain unknown (*Hofmeyer and Treisman, 2009*). One candidate is Ncad, which is required at an early stage of development in both R7 and medulla neurons, but its widespread expression has made it difficult to determine in which neurons it acts to promote R7 synapse stabilization (*Lee et al., 2001*; *Ting et al., 2005*; *Yonekura et al., 2007*; *Özel et al., 2015*).

R7 cells form numerous synapses with Dm8 interneurons, which are essential for ultraviolet spectral preference (*Gao et al., 2008*; *Takemura et al., 2013*) and fall into two classes that are postsynaptic to either yR7 or pR7 cells (*Carrillo et al., 2015*). Each R7 subtype promotes the survival of the class of Dm8 cells (y or pDm8) with which it synapses (*Courgeon and Desplan, 2019*; *Menon et al., 2019*). The Dm8 dendrites that remain in the absence of R7 cells still project to the M6 layer (*Courgeon and Desplan, 2019*), but it is not known whether R7 relies on Dm8 for targeting or survival information. Many synapses between R7 and Dm8 also include the projection neurons Tm5a (yR7) or Tm5b (pR7) as a second postsynaptic element (*Gao et al., 2008*; *Takemura et al., 2015*; *Menon et al., 2019*). In addition, Dm9 interneurons are both pre- and postsynaptic to R7 and mediate center-surround inhibition, similarly to horizontal cells in the mammalian retina (*Takemura et al., 2013*; *Takemura et al., 2015*; *Heath et al., 2020*). Our data indicate that the level of Loaf in Tm5a/b and Dm9 is more important for R7 targeting than its level in Dm8, suggesting that these cells may determine the stability of R7 contacts in the M6 layer. However, we cannot rule out the possibility that the drivers we used to express Loaf in Dm8 did not cause a phenotype because the level or timing of expression was not optimal.

### An excess of Loaf in the postsynaptic cells may destabilize R7 connections

Our observation that the absence of Loaf from R7 only causes a phenotype when Loaf is present in its postsynaptic partners implies that Loaf is not essential for R7 targeting. In *loaf* mutants, redundant mechanisms must stabilize R7 terminals in the M6 layer; cell surface protein interactions often only specify a preference for one synaptic partner over another (*Xu et al., 2019*). Synaptic connections may not form entirely normally in these conditions, as *loaf* mutants show a reduced sensitivity to ultraviolet light when compared to isogenic controls (C.-H. Lee, pers. comm.). Importantly, R7 mistargeting is much more striking when *loaf* is absent from photoreceptors, but present in the

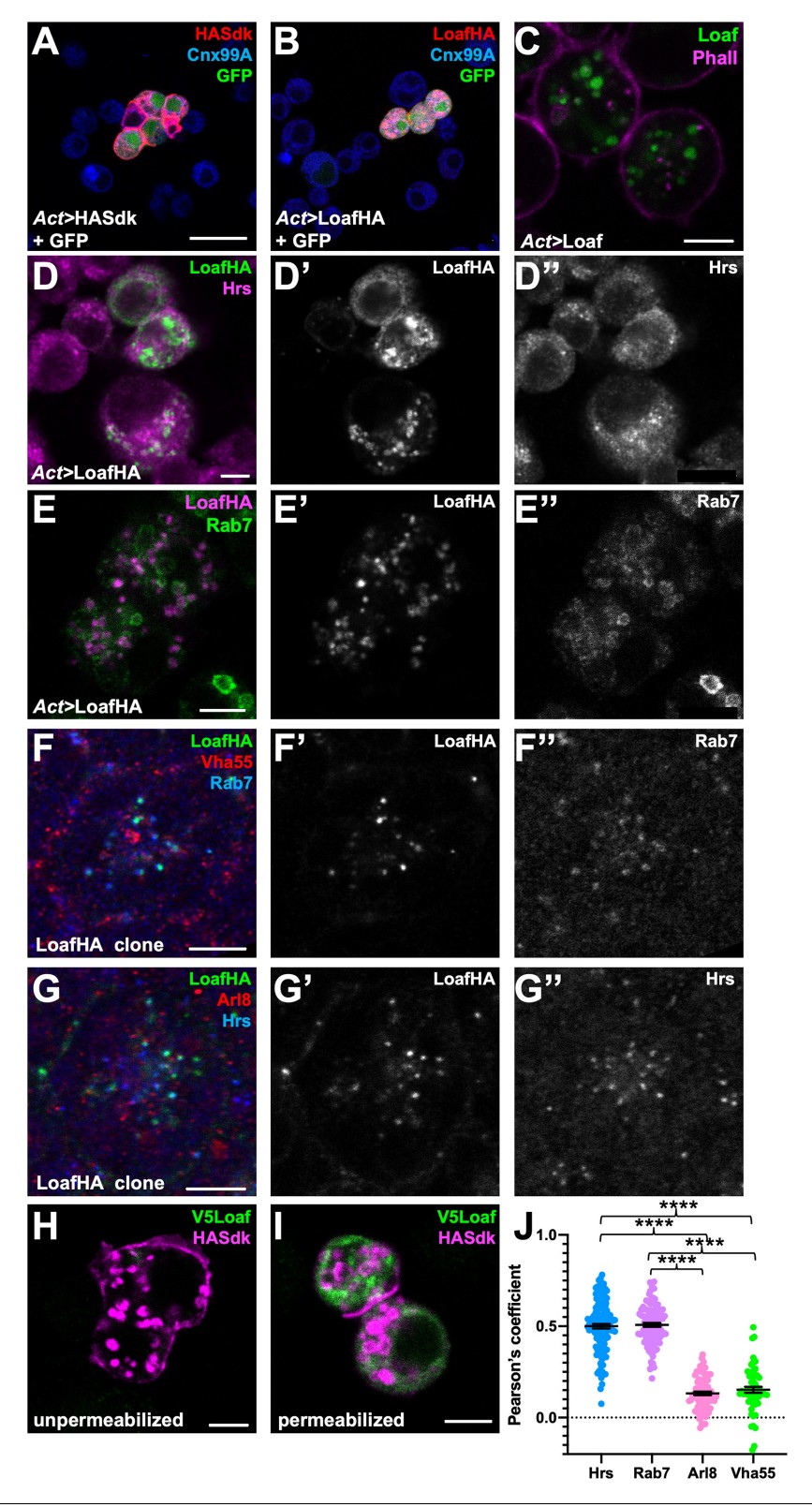

**Figure 6.** Loaf localizes to endosomes. (**A, B**) S2 cells transfected with *Act-GAL4*, *UAS-GFP*, and *UAS-HASdk* (**A**) or *UAS-LoafHA* (**B**) and allowed to aggregate, stained for GFP (green), HA (red) and the ER marker Calnexin 99A (Cnx99A, blue). Sdk localizes to cell contacts and induces aggregation, but Loaf does not. (**C**) S2 cells transfected with *Act-GAL4* and *UAS-Loaf*, stained with anti-Loaf (green) and Phalloidin (magenta). (**D, E**) S2 cells transfected with *Act-GAL4* and *UAS-LoafHA*, stained for HA (D', E', green in D, magenta in E), Hrs (D'', magenta in D), or Rab7 (E'', green in E). Loaf
*Figure 6 continued on next page*

*Figure 6 continued*

localizes to intracellular vesicles that show some colocalization with Hrs and Rab7. (**F, G**) Ommatidia from 42 hr APF pupal retinas in clones expressing UAS-LoafHA, stained for HA (F', G', green in F, G) Rab7 (F'', blue in F), Vha55 (red in F), Hrs (G'', blue in G), and Arl8 (red in G). Loaf colocalizes with the endosomal markers Rab7 and Hrs, but not the lysosomal markers Vha55 and Arl8, in photoreceptors in vivo. (**H, I**) S2 cells transfected with *Act-GAL4*, *UAS-HASdk*, and *UAS-V5Loaf* and incubated with antibodies to HA (magenta) and V5 (green) at room temperature prior to fixation (**H**) or after fixation and permeabilization (**I**). Sdk is detected on the cell surface and in internalized vesicles without fixation, but Loaf is not. (**J**) Quantification of the colocalization of LoafHA with Hrs, Rab7, Arl8, and Vha55 in 42 hr APF retinas by Pearson's correlation. n = 131 ommatidia from 19 retinas (Hrs), 100 ommatidia from 19 retinas (Rab7), 85 ommatidia from 16 retinas (Arl8) or 59 ommatidia from 11 retinas (Vha55). Error bars show mean ± SEM. ****p<0.0001. Scale bars, 20 μm (**A**) or 5 μm (**C–I**).

The online version of this article includes the following source data and figure supplement(s) for figure 6:

**Source data 1.** Data shown in *Figure 6J*.
**Figure supplement 1.** Loaf localizes to endosomes and does not mediate cell aggregation.
**Figure supplement 1—source data 1.** Data shown in *Figure 6—figure supplement 1A*.

brain. We were able to reproduce this mistargeting by expressing *loaf* only in subsets of neurons in the brain that include the major postsynaptic partners of R7. The most parsimonious explanation for these phenotypes is that a mismatch in Loaf expression between R7 and its partners results in mistargeting. A similar phenomenon was observed for the homophilic cell adhesion molecule Klingon, which affects synapse formation when removed from either R7 or glial cells, but not when removed from both simultaneously (*Shimozono et al., 2019*). Matching pre- and postsynaptic levels are also important for the *Drosophila* Teneurin proteins to promote synapse formation (*Hong et al., 2012*; *Mosca et al., 2012*). This type of level matching, in which the presence of a protein in only one of the two partners is more deleterious than its absence from both, is well suited to refining synaptic specificity by eliminating inappropriate connections.

Interestingly, Loaf matching seems to be asymmetric; R7 mistargeting results if Loaf is absent in R7 and present in the postsynaptic cell, but not if it is absent in the postsynaptic cell and present in R7 (*Figure 5I*). It is possible that matching levels in some way neutralize the activity of Loaf, or of a cell surface molecule regulated by Loaf. An excess of this molecule on the postsynaptic cell might prevent it from initiating or stabilizing synapses with R7, or drive it to preferentially connect with other neurons. However, the asymmetry could also reflect the presence of Loaf in multiple postsynaptic cells; loss of *loaf* from only one cell type may not be sufficient to disrupt R7 targeting.

## Loaf may control the trafficking of a cell surface molecule

Our results suggest that Loaf does not itself act as a cell surface adhesion molecule. When epitope-tagged or untagged forms of Loaf are overexpressed in photoreceptors or cultured cells, they localize to intracellular vesicles that include endosomes and do not appear to reach the cell surface. In addition, they do not induce cell aggregation, further arguing against a homophilic adhesion function. CUB domains are present in a variety of functionally distinct proteins, and are thought to bind protein ligands, sometimes in combination with calcium ions (*Gaboriaud et al., 2011*). Some CUB domain proteins are involved in endocytosis of other molecules (*Moestrup and Verroust, 2001*; *Xu and Wang, 2016*), while members of the Neuropilin and Tolloid-like (Neto) family of CUB-LDL proteins are required for the normal localization and activity of glutamate receptors and other postsynaptic proteins (*Zheng et al., 2004*; *Kim et al., 2012*; *Ramos et al., 2015*; *Sheng et al., 2015*). It is thus possible that Loaf controls the level of other proteins on the cell surface by mediating their trafficking or endocytosis. The endosomal protein Commissureless functions in this manner, by trafficking the Roundabout axon guidance receptor directly from the Golgi to endosomes so that it does not reach the cell surface (*Keleman et al., 2002*). In another example, Rab6 and its activator Rich traffic Ncad to the cell surface, facilitating R7 targeting (*Tong et al., 2011*). Differences in Neurexin levels between axons and dendrites are also dependent on endocytosis and sorting (*Ribeiro et al., 2019*), and trafficking of synaptic adhesion molecules in general is highly regulated (*Ribeiro et al., 2018*).

Consistent with this model, we found that loss or gain of Loaf affects the function of coexpressed Lrp4, a presynaptic organizer in the olfactory system that has postsynaptic functions at mammalian neuromuscular junctions (*Yumoto et al., 2012*; *Mosca et al., 2017*). However, Lrp4 alone cannot explain the effects of Loaf, as removing *loaf* from photoreceptors still affects R7 targeting in an *Lrp4*

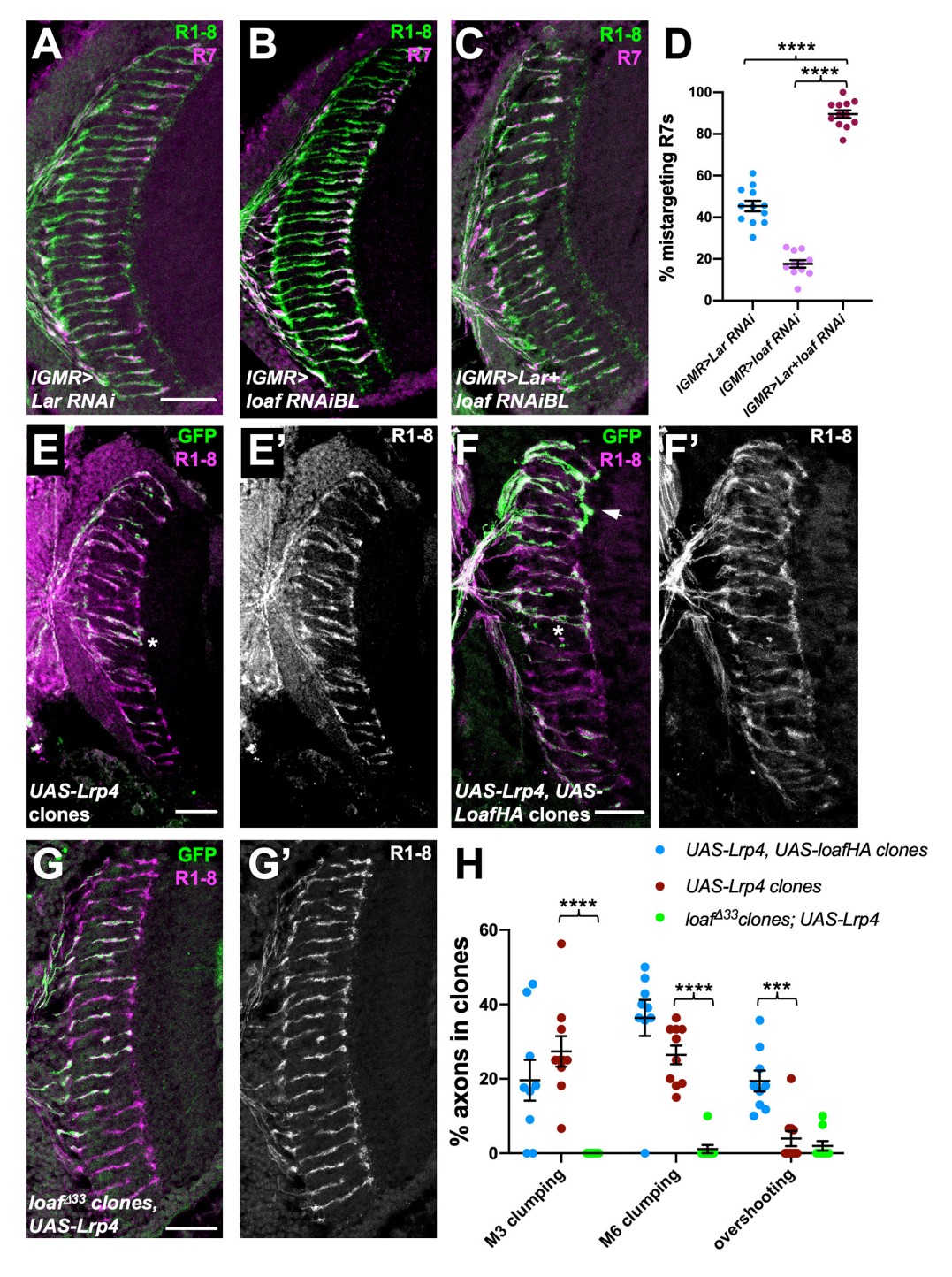

**Figure 7.** *loaf* genetically interacts with *Lar* and *Lrp4*. (A–C) Cryostat sections of adult heads stained for Chp (green) and *panR7-lacZ* (magenta). (A) *lGMR-GAL4, UAS-dcr2/UAS-Lar RNAi*; (B) *lGMR-GAL4, UAS-dcr2; UAS-loaf RNAiBL*; (C) *lGMR-GAL4, UAS-dcr2/UAS-Lar RNAi; UAS-loaf RNAiBL*. With this driver, *Lar* RNAi induces moderate and *loaf* RNAi mild R7 mistargeting, but the combination has a severe phenotype. (D) Quantification of the percentage of R7 axons that failed to reach the M6 layer in the indicated genotypes. n = 12 (*Lar RNAi*, *Lar RNAi +loaf RNAi*) or 11 (*loaf RNAi*). ****, p<0.0001 by unpaired t-test. (E–G) Cryostat sections of adult heads with clones positively labeled with GFP, stained for Chp (E', F', G', magenta in E-G) and GFP (green). (E) Clones expressing *UAS-Lrp4* with *lGMR-GAL4*. (F) Clones expressing *UAS-Lrp4* and *UAS-LoafHA* with *lGMR-GAL4*. (C) *loaf^{Δ33}* clones in which *UAS-Lrp4* is expressed with *lGMR-GAL4*. (H) Quantification of the percentage of labeled R7s of each genotype that show R7 axons clumping together in the M3 (asterisk in F) or M6 (asterisk in E) layers or overshooting the M6 layer (arrow in F). n = 9 heads (*UAS-Lrp4, UASLoafHA*;

*Figure 7 continued on next page*

*Figure 7 continued*

*loaf*^Δ33 clones, *UAS-Lrp4*) or 10 (*UAS-Lrp4*). Error bars show mean ± SEM. ***p<0.001; ****p<0.0001 by multiple t-tests with two-stage linear step-up procedure. Scale bars, 20 μm.

The online version of this article includes the following source data and figure supplement(s) for figure 7:

**Source data 1.** Data shown in *Figure 7D and H*.

**Figure supplement 1.** Lar overexpression can compensate for loss of Loaf.

**Figure supplement 1—source data 1.** Data shown in *Figure 7—figure supplement 1D and J*.

**Figure supplement 2.** Lrp4 is not the only effector of Loaf.

---

null mutant. Loaf may act through a protein similar to Lrp4, or through a combination of proteins. Alternatively, it is possible that under some conditions, perhaps in the presence of other interacting proteins, Loaf itself can reach the cell surface and function there. Some synaptic organizing molecules are transported to axons in lysosome-related vesicles and secreted in a regulated manner (*Arantes and Andrews, 2006*; *Vukoja et al., 2018*; *Ibata et al., 2019*). Further study of the mechanism of Loaf action will provide insight into the cellular mechanisms that enable synaptic connections to be stabilized only on the appropriate cells as neural circuits develop.

## Note added in proof

Further studies by the authors have revealed that *GMR9D03-GAL4* is also expressed in other transmedullary neurons such as Tm15 and Tm25, raising the possibility that Loaf in these neurons could contribute to R7 targeting.

## Materials and methods

**Key resources table**

| Reagent type (species) or resource | Designation | Source or reference | Identifiers | Additional information |
|---|---|---|---|---|
| Gene (*Drosophila melanogaster*) | *loaf* | Flybase | FLYB: FBgn0036202 | *CG6024* |
| Genetic reagent (*D. melanogaster*) | *Rh5-GFP* | Bloomington *Drosophila* Stock Center | BDSC:8600 FLYB: FBti0038634 | FlyBase Symbol: P{Rh5-EGFP.P} |
| Genetic reagent (*D. melanogaster*) | *Rh6-GFP* | Bloomington *Drosophila* Stock Center | BDSC:7461 FLYB: FBti0038637 | FlyBase Symbol: P{Rh6-EGFP.P} |
| Genetic reagent (*D. melanogaster*) | *gl-lacZ* | *Moses and Rubin, 1991* | FLYB: FBtp0001226 | FlyBase Symbol: Ecol \lacZ^5xglBS.38-1 |
| Genetic reagent (*D. melanogaster*) | *R22E09-LexA* | *Pecot et al., 2013* | FLYB: FBtp0108724 | FlyBase Symbol: P{R22E09-nlsLexA::GADfl} |
| Genetic reagent (*D. melanogaster*) | *LexAop:myrTomato* | *Pecot et al., 2013* | FLYB: FBtp0141256 | FlyBase Symbol: M {13xlexAop-tdTomato.IVS.Myr} |
| Genetic reagent (*D. melanogaster*) | *GMR-GAL4* | *Pecot et al., 2013* | FLYB: FBtp0001315 | FlyBase Symbol: P{GAL4-ninaE.GMR} |
| Genetic reagent (*D. melanogaster*) | *ey3.5-FLP* | Bloomington *Drosophila* Stock Center | BDSC:35542 FLYB: FBti0141243 | FlyBase Symbol: P{ey3.5-FLP.B} |
| Genetic reagent (*D. melanogaster*) | *Act>CD2>GAL4* | Bloomington *Drosophila* Stock Center | BDSC:4780 FLYB: FBti0012408 | FlyBase Symbol: P{GAL4-Act5C(FRT.CD2).P}S |
| Genetic reagent (*D. melanogaster*) | *lGMR-GAL4* | Bloomington *Drosophila* Stock Center | BDSC: 8605 FLYB: FBti0058798 | FlyBase Symbol: P {longGMR-GAL4} |
| Genetic reagent (*D. melanogaster*) | *UAS-loaf RNAiBL* | Bloomington *Drosophila* Stock Center | BDSC: 28625 FLYB: FBti0127178 | FlyBase Symbol: RNAiBL P {TRiP.JF03040}attP2 |
| Genetic reagent (*D. melanogaster*) | *UAS-loaf RNAiKK* | Viennna *Drosophila* Resource Center | VDRC: 102704 FLYB: FBst0474570 | FlyBase Symbol: P {KK112220}VIE-260B |
| Genetic reagent (*D. melanogaster*) | *UAS-LarRNAi* | Viennna *Drosophila* Resource Center | VDRC: 107996 FLYB: FBst0479809 | FlyBase Symbol: P {KK100581}VIE-260B |

*Continued on next page*

*Continued*

| Reagent type (species) or resource | Designation | Source or reference | Identifiers | Additional information |
|---|---|---|---|---|
| Genetic reagent (*D. melanogaster*) | *UAS-dcr2* | Bloomington *Drosophila* Stock Center | BDSC: 24650 FLYB: FBti0100275 | FlyBase Symbol: P{UAS-Dcr-2.D} |
| Genetic reagent (*D. melanogaster*) | *panR7-lacZ* | *Hofmeyer et al., 2006* | FLYB: FBtp0022109 | FlyBase Symbol: P{PanR7-lacZ} |
| Genetic reagent (*D. melanogaster*) | *nos-Cas9* | Bloomington *Drosophila* Stock Center | BDSC: 54591 FLYB: FBti0159183 | FlyBase Symbol: M{nos-Cas9.P}ZH-2A |
| Genetic reagent (*D. melanogaster*) | *UAS-Cas9-P2* | Bloomington *Drosophila* Stock Center | BDSC: 58986 FLYB: FBti0166500 | FlyBase Symbol: P{UAS-Cas9.P2}attP2 |
| Genetic reagent (*D. melanogaster*) | *DIP-γ-GAL4* | *Carrillo et al., 2015* | FLYB: FBal0319064 | FlyBase Symbol: DIP-γ$^{MI03222-GAL4}$ |
| Genetic reagent (*D. melanogaster*) | *tj-GAL4$^{NP1624}$* | Kyoto *Drosophila* Stock Center | Kyoto: 104055 FLYB: FBst0302922 | FlyBase Symbol: P{GawB} NP1624/CyO |
| Genetic reagent (*D. melanogaster*) | *drf-GAL4* | *Brody et al., 2012* | FLYB: FBal0270054 | FlyBase Symbol: GAL4$^{vvl.43}$ |
| Genetic reagent (*D. melanogaster*) | *loaf$^{MiMIC-GFSTF}$* | Bloomington *Drosophila* Stock Center | BDSC: 64464 FLYB: FBti0181845 | FlyBase Symbol: Mi{PT-GFSTF.1}CG6024$^{MI00316-GFSTF.1}$ |
| Genetic reagent (*D. melanogaster*) | *ap-GAL4* | Bloomington *Drosophila* Stock Center | BDSC: 3041 FLYB: FBti0002785 | FlyBase Symbol: P{GawB} ap$^{md544}$ |
| Genetic reagent (*D. melanogaster*) | *ChAT-GAL4* | Bloomington *Drosophila* Stock Center | BDSC: 6798 FLYB: FBti0024050 | FlyBase Symbol: P{ChAT-GAL4.7.4} |
| Genetic reagent (*D. melanogaster*) | *repo-GAL4* | Bloomington *Drosophila* Stock Center | BDSC: 7415 FLYB: FBti0018692 | FlyBase Symbol: P{GAL4} repo |
| Genetic reagent (*D. melanogaster*) | *hth-GAL4* | *Wernet et al., 2003* | FLYB: FBti0058519 | FlyBase Symbol: P{GawB} hth$^{GAL4}$ |
| Genetic reagent (*D. melanogaster*) | *bsh-GAL4* | *Hasegawa et al., 2011* | FLYB: FBtp0069756 | FlyBase Symbol: P{bsh-GAL4.H} |
| Genetic reagent (*D. melanogaster*) | *Vsx-GAL4* | Bloomington *Drosophila* Stock Center | BDSC: 29031 FLYB: FBti0037957 | FlyBase Symbol: P{GawB} MzVum |
| Genetic reagent (*D. melanogaster*) | *VGlut-GAL4* | Bloomington *Drosophila* Stock Center | BDSC: 26160 FLYB: FBti0076967 | FlyBase Symbol: P{GawB} VGlut$^{OK371}$ |
| Genetic reagent (*D. melanogaster*) | *GMR9D03-GAL4* | Bloomington *Drosophila* Stock Center | BDSC: 40726 FLYB: FBti0152068 | FlyBase Symbol: P {GMR9D03-GAL4}attP2 |
| Genetic reagent (*D. melanogaster*) | *GH146-GAL4* | Bloomington *Drosophila* Stock Center | BDSC: 30026 FLYB: FBti0016783 | FlyBase Symbol: P{GawB} GH146 |
| Genetic reagent (*D. melanogaster*) | *dve$^{NP3428}$-GAL4* | Kyoto *Drosophila* Stock Center | Kyoto: 113273 FLYB: FBti0035416 | FlyBase Symbol: P{GawB} dve$^{NP3428}$ |
| Genetic reagent (*D. melanogaster*) | *vg-GAL4* | Bloomington *Drosophila* Stock Center | BDSC: 6819 FLYB: FBal0047077 | FlyBase Symbol: GAL4$^{vg.PM}$ |
| Genetic reagent (*D. melanogaster*) | *ort$^{C1a}$-GAL4* | Bloomington *Drosophila* Stock Center | BDSC: 56519 FLYB: FBti0161257 | FlyBase Symbol: P{ort-GAL4.C1a} |
| Genetic reagent (*D. melanogaster*) | *ort$^{C2b}$-GAL4* | *Ting et al., 2014* | FLYB: FBtp0093983 | FlyBase Symbol: P{ort-GAL4.C2b} |
| Genetic reagent (*D. melanogaster*) | *GMR64H01-GAL4* | Bloomington *Drosophila* Stock Center | BDSC: 39322 FLYB: FBti0137495 | FlyBase Symbol: P {GMR64H01-GAL4}attP2 |
| Genetic reagent (*D. melanogaster*) | *UAS-LarHA* | *Hofmeyer and Treisman, 2009* | FLYB: FBal0193546 | FlyBase Symbol: Lar$^{UAS.Tag:HA}$ |
| Genetic reagent (*D. melanogaster*) | *UAS-Lrp4HA* | *Mosca et al., 2017* | FLYB: FBal0326704 | FlyBase Symbol: Lrp4$^{UAS.Tag:HA,Tag:FLAG}$ |
| Genetic reagent (*D. melanogaster*) | *Lrp4$^{dalek}$* | *Mosca et al., 2017* | FLYB: FBal0326703 | FlyBase Symbol: Lrp4$^{dalek}$ |
| Genetic reagent (*D. melanogaster*) | *Liprin-α$^{oos}$* | *Hofmeyer et al., 2006* | FLYB: FBal0193553 | FlyBase Symbol: Liprin-α$^{oos}$ |

*Continued on next page*

*Continued*

| Reagent type (species) or resource | Designation | Source or reference | Identifiers | Additional information |
|---|---|---|---|---|
| Genetic reagent (*D. melanogaster*) | *GMR9D03-DBD* | Bloomington *Drosophila* Stock Center | BDSC: 68766 FLYB: FBti0192157 | FlyBase Symbol: P{R9D03-GAL4.DBD}attP2 |
| Genetic reagent (*D. melanogaster*) | *GMR38H04-AD* | Bloomington *Drosophila* Stock Center | BDSC: 75758 FLYB: FBti0188278 | FlyBase Symbol: P {R38H04-p65.AD}attP40 |
| Genetic reagent (*D. melanogaster*) | *MCFO-1* | **Nern et al., 2015** | FLYB: FBti0169283 | FlyBase Symbol: PBac {10XUAS(FRT.stop)myr:: smGdP-HA}VK00005-P {10xUAS(FRT.stop)myr:: smGdP-V5-THS-10xUAS (FRT.stop)myr::smGdP-FLAG}su(Hw) |
| Genetic reagent (*D. melanogaster*) | *TSG101[2]* | **Moberg et al., 2005** | FLYB: FBal0212938 | FlyBase Symbol: TSG101[2] |
| Genetic reagent (*D. melanogaster*) | *loaf[Δ33]* | This paper | | CRISPR deletion allele; see Materials and methods |
| Genetic reagent (*D. melanogaster*) | *loaf[Δ20]* | This paper | | CRISPR deletion allele; see Materials and methods |
| Genetic reagent (*D. melanogaster*) | *UAS-LoafHA* | This paper | | Inserted at VK1 attP site |
| Genetic reagent (*D. melanogaster*) | *UAS-Loaf* | This paper | | Inserted at VK1 attP site |
| Genetic reagent (*D. melanogaster*) | *pCFD4-loaf sgRNAs* | This paper | | Inserted at attP40 site |
| Cell line (*D. melanogaster*) | S2 | Laboratory of Ruth Lehmann | FLYB:FBtc0000181; RRID: CVCL_Z992 | FlyBase symbol: S2-DRSC. |
| Antibody | Anti-Chp (Mouse monoclonal) | Developmental Studies Hybridoma Bank | Cat# 24B10, RRID:AB_528161 | IF(1:50) |
| Antibody | Anti-GFP (Chicken polyclonal) | Life Technologies | Cat# A10262, RRID:AB_2534023 | IF(1:400) |
| Antibody | Anti-HA (Rat monoclonal) | Sigma (Roche 3F10) | Cat# 11 867 423 001, RRID:AB_390918 | IF(1:400) |
| Antibody | Anti-β-galactosidase (Rabbit polyclonal) | Fisher | Cat# A11132, RRID:AB_221539 | IF(1:100) |
| Antibody | Anti-Ncad (Rat monoclonal) | Developmental Studies Hybridoma Bank | Cat# DN-Ex #8, RRID:AB_528121 | IF(1:50) |
| Antibody | Anti-DsRed (Rabbit polyclonal) | Takara Bio | Cat# 632496, RRID:AB_10013483 | IF(1:50) |
| Antibody | Anti-Loaf (Guinea pig polyclonal) | Proteintech (this paper) | | IF(1:400) WB(1:1000) See Materials and methods |
| Antibody | Anti-Cnx99A (Mouse monoclonal) | Developmental Studies Hybridoma Bank | Cat# Cnx99A 6-2-1, RRID:AB_2722011 | IF(1:10) |
| Antibody | Anti-Hrs (Mouse monoclonal) | Developmental Studies Hybridoma Bank | Cat# Hrs 27–4, RRID:AB_2618261 | IF(1:10) |
| Antibody | Anti-Rab7 (Mouse monoclonal) | Developmental Studies Hybridoma Bank | Cat# Rab7, RRID:AB_2722471 | IF(1:10) |
| Antibody | Anti-ATP6V1B1 (Rabbit polyclonal) | Abgent | Cat# AP11538C, RRID:AB_10816749 | IF(1:200) |
| Antibody | Anti-Arl8 (Rabbit polyclonal) | Developmental Studies Hybridoma Bank | Cat# Arl8, RRID:AB_2618258 | IF(1:200) |
| Antibody | Anti-Elav (Rat monoclonal) | Developmental Studies Hybridoma Bank | Cat# Rat-Elav-7E8A10 anti-elav, RRID:AB_528218 | IF(1:100) |

*Continued on next page*

*Continued*

| Reagent type (species) or resource | Designation | Source or reference | Identifiers | Additional information |
|---|---|---|---|---|
| Antibody | Anti-Notch (Mouse monoclonal) | Developmental Studies Hybridoma Bank | Cat# C17.9C6, RRID:AB_528410 | IF(1:10) |
| Antibody | Anti-Arm (Mouse monoclonal) | Developmental Studies Hybridoma Bank | Cat# N2 7A1 Armadillo, RRID:AB_528089 | IF(1:10) |
| Antibody | Anti-GFP (Sheep polyclonal) | BioRad | Cat# 4745–1051, RRID:AB_619712 | IF(1:200) |
| Antibody | Anti-RFP (Rabbit polyclonal) | MBL International | Cat# PM005, RRID:AB_591279 | IF(1:500) |
| Antibody | Anti-V5 (Rabbit polyclonal) | Abcam | Cat# ab9116, RRID:AB_307024 | IF(1:1000) |
| Antibody | Anti-FLAG (Mouse monoclonal) | Sigma | Cat# F3165, RRID:AB_259529 | IF(1:500) |
| Antibody | Anti-Dac (Mouse monoclonal) | Developmental Studies Hybridoma Bank | Cat# mAbdac2-3, RRID:AB_528190 | IF(1:40) |
| Antibody | Anti-Bsh (Rabbit polyclonal) | Özel et al., 2021 | | IF(1:1800) |
| Antibody | Anti-Runt (Guinea pig polyclonal) | Genscript (this paper) | | IF(1:600) See Materials and methods |
| Antibody | Anti-β-tubulin (Mouse monoclonal) | Sigma | Cat# T4026, RRID:AB_477577 | WB (1:10,000) |
| Recombinant DNA reagent | UAS-HASdk | Astigarraga et al., 2018 | | In UASt-attB |
| Recombinant DNA reagent | UAS-LoafHA | *Drosophila* Genomics Resource Center | Clone UFO07678 | In UASt-attB |
| Recombinant DNA reagent | UAS-Loaf | This paper | | See Materials and methods |
| Recombinant DNA reagent | UAS-V5Loaf | This paper | | See Materials and methods |
| Sequence-based reagent | Loaf_F | This paper | PCR primer | CGCACGAACTTTGTGACACT |
| Sequence-based reagent | Loaf_R | This paper | PCR primer | CTCAAGTCAATCGGTCCTTCC |
| Commercial assay or kit | SuperSignal WestPico | ThermoFisher | Cat # 34579 | |
| Software, algorithm | Fiji-ImageJ | NIH | https://fiji.sc/ | |

## Fly stocks and genetics

Fly stocks used were *Rh5-GFP* (Bloomington *Drosophila* Stock Center [BDSC] #8600); *Rh6-GFP* (BDSC #7461); *gl-lacZ* (**Moses and Rubin, 1991**), *R22E09-LexA, LexAop-myrTomato; GMR-GAL4* (**Pecot et al., 2013**); *ey3.5-FLP, Act>CD2>GAL4* (BDSC #35542 and #4780); *lGMR-GAL4* (BDSC #8605); *UAS-loaf RNAiBL* P{TRiP.JF03040}attP2 (BDSC #28625); *UAS-loaf RNAiKK* P{KK112220}VIE-260B (Vienna *Drosophila* Resource Center [VDRC] #102704); *UAS-Lar RNAi* P{KK100581}VIE-260B (VDRC #107996); *UAS-dcr2* (BDSC #24650); *panR7-lacZ* (**Hofmeyer et al., 2006**); *nos-Cas9* (BDSC #54591); *UAS-Cas9-P2* (BDSC #58986); *DIP-γ-GAL4* (**Carrillo et al., 2015**); *tj-GAL4^NP1624* (Kyoto Stock Center #104055); *drf-GAL4* (**Brody et al., 2012**); *Mi{PT-GFSTF.1}CG6024^MI00316-GFSTF.1* (BDSC #64464); *ap^md544-GAL4* (BDSC #3041); *ChAT-GAL4* (BDSC #6798); *repo-GAL4* (BDSC #7415); *hth-GAL4* (**Wernet et al., 2003**); *bsh-GAL4* (**Hasegawa et al., 2011**); *Vsx-GAL4* (BDSC #29031); *VGlut-GAL4* (BDSC #26160); *GMR9D03-GAL4* (BDSC #40726); *GH146-GAL4* (BDSC #30026); *dve^NP3428-GAL4* (Kyoto Stock Center #113273); *vg-GAL4* (BDSC #6819); *ort^C1a-GAL4* (BDSC #56519); *ort^C2b-GAL4* (**Ting et al., 2014**); *GMR64H01-GAL4* (BDSC #39322); *UAS-LarHA* (**Hofmeyer and Treisman, 2009**); *UAS-Lrp4HA; Lrp4^dalek* (**Mosca et al., 2017**); *Liprin-α^oos* (**Hofmeyer et al., 2006**); *GMR9D03-DBD* (BDSC #68766); *GMR38H04-AD* (BDSC #75758); *MCFO-1* (**Nern et al., 2015**), and *TSG101^2* (**Moberg et al., 2005**). *loaf* mutant clones and *loaf* mutant clones overexpressing other proteins were generated using *ey3.5-FLP, UAS-CD8GFP; lGMR-GAL4; FRT80, tub-GAL80*. *loaf^MiMIC-GFSTF* clones were generated using *eyFLP; lGMR-GAL4, UAS-myr-tdTomato; FRT80, tub-GAL80*. Clones in

which specific cell types were labeled were generated by crossing *ort^{C2b}*-GAL4 (or other GAL4 lines); *FRT80* (or *FRT80, loaf^{Δ33}*) to *hs-FLP122, UAS-CD8GFP; FRT80, tub-GAL80*. Overexpression clones were generated by crossing *UAS-LoafHA* (or *UAS-Loaf, UAS-LarHA* or *UAS-Lrp4HA*); *FRT82* to *ey3.5-FLP, UAS-CD8GFP; lGMR-GAL4; FRT82, tub-GAL80*. To obtain sparse labeling of Tm5a/b/c neurons, flies with the genotype *hsflp2PEST; UAS>stop>CD4-tdGFP/CyO; GMR9D03-GAL4* were heat-shocked for 7 min at late L3 stage and dissected in the adult. *loaf* mutant clones in a background of Lrp4 overexpression were generated by crossing *UAS-Lrp4HA; lGMR-GAL4, FRT80, loaf^{Δ33}* to *eyFLP; FRT80, ubi-RFP*. Lrp4 mutant clones were generated using *ey-FLP, tub-GAL80, FRT19; lGMR-GAL4, UAS-CD8-GFP*. To restore Loaf to specific cell types in a *loaf* mutant background, *tj-GAL4* (or other GAL4 lines); *loaf^{Δ33}/SM6-TM6B* was crossed to *UAS-LoafHA, UAS-myrTomato; loaf^{Δ33}/SM6-TM6B* or to *UAS-LoafHA; panR7-lacZ, loaf^{Δ33}/SM6-TM6B*. GAL4 lines on the third chromosome were recombined with *loaf^{Δ33}* and recombinants carrying *loaf* were identified by PCR, except for GAL4 lines inserted at the *attP2* site, which is very close to *loaf*. In these cases, new *loaf* alleles were directly introduced by CRISPR onto the GAL4 chromosome using *nos-Cas9* and our transgenic *loaf sgRNA* flies, and identified by PCR.

## Immunohistochemistry

Adult heads were dissected in cold 0.1 M sodium phosphate buffer (PB) pH 7.4, fixed in 4% formaldehyde in PB for 4 hr at 4°C and washed in PB. Heads were then submerged in a sucrose gradient (5%, 10%, 20%) and left in 25% sucrose overnight at 4°C for cryoprotection. Heads were embedded in OCT tissue freezing medium and frozen in dry ice/ethanol, and 12 μm sections were cut on a cryostat. Sections were post-fixed in 0.5% formaldehyde in PB for 30 min at room temperature and washed three times in PB with 0.1% Triton (PBT) before incubation with primary antibodies overnight at 4°C. Sections were washed four times for 20 min with PBT and incubated with secondary antibodies for 2 hr at room temperature. Sections were washed again four times for 20 min before mounting in Fluoromount-G (Southern Biotech).

Pupal and adult whole brains were fixed in 4% paraformaldehyde in PBS for 30 min at room temperature, washed 3 times for 10 min in PBST (PBS + 0.4% Triton-X 100) and blocked in PBST +10% donkey serum prior to incubation with primary antibodies overnight at 4°C. For Loaf staining of pupal brains, this incubation was extended to 4 days. Samples were washed in PBST three times for at least 1 hr each and incubated with secondary antibodies for 2.5 hr at room temperature. Samples were washed three times for 20 min in PBST and once in PBS before mounting in SlowFade Gold AntiFade reagent (Life Technologies) on bridge slides. Pupal retinas were fixed in 4% paraformaldehyde in PBS for 30 min on ice, washed for 15 min in PBT and incubated with primary antibodies overnight at 4°C. Retinas were washed three times for 5 min with PBT, incubated with secondary antibodies for 2 hr at 4°C and washed again three times for 5 min before mounting in 80% glycerol in PBS. Confocal images were collected with Leica SP8 and Zeiss LSM510 confocal microscopes.

The primary antibodies used were mouse anti-Chp (1:50; Developmental Studies Hybridoma Bank [DSHB] 24B10), chicken anti-GFP (1:400; Life Technologies), rat anti-HA (1:50; Roche 3F10), rabbit anti-β galactosidase (1:100, Fisher), rat anti-Ncad (1:50; DSHB), rabbit anti-dsRed (1:500; Takara Bio), guinea pig anti-Loaf (1:400, Proteintech), mouse anti-Cnx99A (1:10, DSHB 6-2-1), mouse anti-Hrs (1:10, DSHB 27–4), mouse anti-Rab7 (1:10, DSHB), rabbit anti-ATP6V1B1 (Vha55; 1:200, Abgent), rabbit anti-Arl8 (1:200; DSHB), rat anti-Elav (1:100, DSHB), mouse anti-Notch (1:10; DSHB C17.9C6), mouse anti-Arm (1:10; DSHB N2 7A1), sheep anti-GFP (1:200, Bio-Rad #4745–1051), rabbit anti-RFP (1:500; MBL International #PM005), rabbit anti-V5 (1:1000; Abcam ab9116), mouse anti-FLAG (1:500, Sigma F3165), mouse anti-Dac (1:40; DSHB mAbdac2-3), rabbit anti-Bsh (1:1800) (Özel et al., 2021), and guinea pig anti-Runt (1:600; GenScript). Rhodamine-phalloidin (Invitrogen R415) was used at 1:20. The Loaf polyclonal antibody was made by Proteintech using the cytoplasmic domain (aa 292–378) as an antigen. Guinea pig anti-serum was affinity purified. Guinea pig anti-Runt was made by GenScript using the full-length protein as an antigen. Secondary antibodies (Jackson Immunoresearch and Life Technologies) were coupled to the fluorochromes Cy3, AlexaFluor 488, or AlexaFluor 647.

## Quantifications

To quantify the R7 targeting defect, fluorescent image stacks of 12 µm adult head sections labeled for *gl-lacZ*, *Rh3/4-lacZ*, or anti-24B10 were gathered in 0.5 µm steps. Maximum intensity projections were obtained and termini projecting beyond the R8 layer were counted as 'R7 correctly targeted' and those stopping in the R8 layer were counted as 'R7 incorrectly targeted.' Termini in the R8 layer were counted as total cartridge number per section. The percentage of mistargeting R7s was calculated for each section, except that when the phenotype was scored in photoreceptor clones, the percentage was calculated from all mistargeting R7 axons within clones from all sections. To quantify defects in UAS-LRP4 clones, GFP-labeled R7 termini that contacted each other in the M3 layer were counted as 'M3 clumping', while termini hyperfasciculating in the M6 layer were counted as 'M6 clumping.' Three termini contacting each other were counted as two instances of M3 or M6 clumping depending on in which layer the clumps resided. GFP-labeled R7 termini that extended past the M6 layer were counted as 'overshooting.' A terminus that extended past the M6 layer and turned to contact another clone was counted both as 'M6 clumping' and 'overshooting.' The percentage of R7s belonging to each of these groups was calculated from all the R7 clones within each section.

To quantify cell aggregates in S2 cell culture experiments, fluorescent image stacks from fixed cells that had been labeled for GFP and HA were gathered in 0.5 µm steps. Each image was examined for GFP positive (control) or GFP and HA-positive (Sdk or Loaf) cells that contacted each other as aggregates. The number of cells in each aggregate was counted for each image. To measure intracellular colocalization, single confocal slices were processed with a median filter with neighborhood of 1 in ImageJ and each channel was linear contrast enhanced to spread values evenly from 0 to 255. A rectangular ROI was drawn around the central region of a single ommatidium and ImageJ was used to calculate Pearson's correlation coefficient on each region with pixel intensity above a threshold of 16 out of a range of 255, to eliminate background.

## Western blotting

To extract proteins, adult heads were dissected and frozen on dry ice, and then homogenized in Laemmli buffer (4% SDS, 20% glycerol, 120 mM Tris-Cl pH 6.8, 0.02% bromophenol blue, 10% beta-mercaptoethanol). Samples were heated at 95°C for 5 min and loaded onto a SDS-PAGE gel. Gels were run first at 80 volts for 20 min, then 100 volts for the remainder of the time and transferred onto nitrocellulose membranes (Bio-Rad) for one hour at 100 volts. Membranes were washed for 5 min in TBST (20 mM Tris (pH 7.6), 136 mM NaCl, 0.2% Tween-20), and blocked in 5% low-fat milk in TBST solution for one hour. Membranes were incubated overnight with primary antibody in TBST with 5% milk at 4°C, washed three times for 10 min in TBST and incubated in horseradish peroxidase-conjugated secondary antibodies (1:10,000; Jackson ImmunoResearch) at room temperature in TBST with 5% milk for 2 hr. Membranes were washed three times for 10 min in TBST and once for 10 min in TBS. Enhanced chemiluminescence (Thermo SuperSignal WestPico) was used to develop the blots. Primary antibodies used were guinea pig anti-Loaf (1:1000, Proteintech) and mouse anti ß-tubulin (1:10,000; Sigma, T4026).

## Cloning and transgenic lines

UAS-Loaf-FLAG-HA is clone UFO07678 (*Drosophila* Genomics Resource Center). UAS-Loaf was cloned by inserting an Nhe I/Xba I fragment of clone UFO07678 into the Xba I site of pUAST-attB. Both constructs were integrated into the VK1 PhiC31 site at position 59D3. The *loaf* sgRNA sequences GCTGGTGATTACGTCGGTGA (*loaf* gRNA 1) and TGCGGGACCATCCGGGTACC (*loaf* gRNA 2) identified on http://www.flyrnai.org/crispr2 were made with gene synthesis in pUC57 (GenScript) and cloned into pCFD4 (*Port et al., 2014*) by Gibson assembly. The construct was integrated into the attP40 site at 25C6. These flies were crossed to *nos*-Cas9 flies to make germline mosaic flies. The progeny of these flies were crossed to balancer flies and screened by PCR using primers outside the expected deletion (CGCACGAACTTTGTGACACT and CTCAAGTCAATCGGTCCTTCC). In *loaf*[Δ20], the deletion extends from TACGTCGGTGA in gRNA1 through TGCGGG in sgRNA2, creating a frameshift and a stop codon after 30 novel amino acids. *loaf*[Δ33] has the final CGGTGA of sgRNA replaced by GATT, and then deletes through TGCGGG in sgRNA2, creating a stop codon immediately following Thr 208 at the end of the CUB domain. Injections and screening of transgenic flies were carried out by Genetivision. A V5-Loaf construct in which the V5 epitope tag

(GKPIPNPLLGLDST) was inserted following H90, four residues after the predicted signal peptide cleavage site, was synthesized by GenScript and cloned into pUASTattB using the EcoRI and XbaI sites.

## S2 cell culture and aggregation assay

S2 cells were grown in Schneider's *Drosophila* Medium (GIBCO Invitrogen) with 10% heat inactivated fetal bovine serum and 50 units/ml penicillin-50 g/ml streptomycin (GIBCO Invitrogen) at 25°C. Cells were spun down and resuspended in PBS. Poly-L-lysine-treated slides were covered with 0.1–0.2 ml of the cell suspension. Cells were fixed for 10 min at room temperature with 4% paraformaldehyde, permeabilized for 15 min with 0.2% PBT, then blocked with 10% normal donkey serum. Slides were incubated with primary antibodies overnight at 4°C in a humid chamber, washed four times with PBS, and incubated with secondary antibody at room temperature for 1–2 hr. Samples were washed three times with PBS before mounting with Vectashield (Vector Labs). To stain cell surface proteins, cells were incubated with primary antibody in PBS for 2 hr at room temperature prior to fixation. Pictures were collected on a Leica SP8 confocal microscope.

For aggregation assays, S2 cells were pelleted 48 hr after transient transfection using Effectene Transfection Reagent (Qiagen) and washed in fresh medium. A total of 2.5 ml of cells at a concentration of $4 \times 10^6$ cells/ml were rocked at 50 rpm for at least 3 hr. Plates were then analyzed for the presence of cell aggregates. Pictures were collected on a Zeiss AxioZoom microscope.

## Acknowledgements

We thank Steve Cohen, Max Courgeon, Chi-Hon Lee, Ken Moberg, Tim Mosca, Larry Zipursky, the Bloomington *Drosophila* Stock Center, the Vienna *Drosophila* Resource Center, the Kyoto Stock Center, the *Drosophila* Genomics Resource Center and the Developmental Studies Hybridoma Bank for fly stocks and reagents, and Flybase for invaluable information. We thank Michael Cammer of the NYU Langone Microscopy Laboratory, which is supported by grant P30CA016087, for help with quantifying colocalization. We are grateful to Justine Oyallon for her contributions to the early stages of the project, and to Hui Hua Liu and DanQing He for technical assistance. The manuscript was improved by the critical comments of Hongsu Wang. This work was supported by NIH grants R01GM089799 and R01NS112211 to J.E.T. and by fellowship F31EY025568 to J.D. I.H. was supported by a Human Frontier Science Program postdoctoral fellowship (LT000757/2017).

## Additional information

### Funding

| Funder | Grant reference number | Author |
| --- | --- | --- |
| National Institutes of Health | R01GM089799 | Jessica E Treisman |
| National Institutes of Health | R01NS112211 | Jessica E Treisman |
| National Institutes of Health | F31EY025568 | Jessica Douthit |
| Human Frontier Science Program | LT000757/2017 | Isabel Holguera |

The funders had no role in study design, data collection and interpretation, or the decision to submit the work for publication.

### Author contributions

Jessica Douthit, Conceptualization, Data curation, Formal analysis, Investigation, Methodology, Writing - review and editing; Ariel Hairston, Conceptualization, Data curation, Formal analysis, Validation, Investigation, Writing - review and editing; Gina Lee, Carolyn A Morrison, Isabel Holguera, Investigation, Writing - review and editing; Jessica E Treisman, Conceptualization, Data curation, Supervision, Funding acquisition, Investigation, Writing - original draft, Project administration, Writing - review and editing

Author ORCIDs
Isabel Holguera https://orcid.org/0000-0003-2796-6596
Jessica E Treisman https://orcid.org/0000-0002-7453-107X

Decision letter and Author response
Decision letter https://doi.org/10.7554/eLife.65895.sa1
Author response https://doi.org/10.7554/eLife.65895.sa2

## Additional files

Supplementary files
• Transparent reporting form

Data availability

All data generated or analyzed during this study are included in the manuscript and supporting files.

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
