## [Decision Letter]

**Acceptance summary:**

Your study on the requirement for the CUB-LDL protein Lost and found (Loaf) in the *Drosophila* R7 photoreceptor interactions with its target neurons sheds light on our understanding of axon-target cell matching via cell surface proteins, indeed, poorly understood in all species. You demonstrate that when Loaf is absent in R7, the presence of Loaf in synaptic partner neurons is in fact detrimental to R7 targeting. Your precise analyses and model indicate that proper matching depends on the relative levels of Loaf expression in R7 and its synaptic partners. The reviewers were pleased with your amendments and found the results exciting.

**Decision letter after peer review:**

[Editors’ note: the authors submitted for reconsideration following the decision after peer review. What follows is the decision letter after the first round of review.]

Thank you for submitting your work entitled "R7 photoreceptor axon targeting requires matching levels of the lost and found protein in R7 and its synaptic partners" for consideration by *eLife*. Your article has been reviewed by 3 peer reviewers, and the evaluation has been overseen by a Reviewing Editor and a Senior Editor. The following individual involved in review of your submission has agreed to reveal their identity: Takashi Suzuki (Reviewer #3).

Our decision has been reached after consultation between the reviewers. Based on these discussions and the individual reviews below, we regret to inform you that your work will not be considered for publication in *eLife* at this time.

Your study pointing to trafficking of a putative transmembrane protein, to attain matched levels of this protein on R7 and its synaptic partners, in mediating stable synapse formation, was viewed overall favorably by the reviewers. This study could be valuable because although cell surface proteins are thought to mediate cell-cell recognition, few have detected these proteins, or addressed whether quantitative comparisons of the levels of such proteins could mediate synaptic specificity.

Nonetheless, as you can see from the reviews below, and comments in the reviewer consultation session, we thought that while the work is promising, it will require additional data and significant effort before the suggested model is substantiated. Several controls are missing to support the matched levels/competition model, and developmental analysis of expression patterns, levels and mistargeting phenotypes should be included to bolster your arguments. In addition, the language around synaptogenesis needs to be changed unless direct evidence related to synapse development is provided.

Therefore, while we consider the work to have potential, essential additional data is required to support the central claims of the paper. Thus, we request revisions (without a deadline) and encourage you to post the paper to a preprint server along with the reviews from *eLife*.

*Reviewer #1:*

In this manuscript the authors focus on the role of Lost and found (Loaf), a CUB-LDL domain protein in R7 axon targeting. The context-specific nature of whether Loaf LOF in photoreceptors causes an R7 mistargeting effect is an interesting result and the authors have proposed a conceptually interesting model to explain the data, i.e. Quantitative comparisons of the levels of Loaf between R7s and their synaptic partners determine appropriate R7 targeting when Loaf levels are matched. However, while the data presented in this manuscript is consistent with the model, the model itself has not been tested sufficiently. Overall, I believe the work is interesting and has potential but requires essential additional data to support the central claims of the paper.

1. The authors suggest that the R7 mistargeting phenotype is the consequence of lower loaf levels in R7s relative to R7 synaptic partners (Tm5a/b and/or Dm9) since removing Loaf from photoreceptors alone results in mistargeting and likewise restoring Loaf in synaptic partners of R7 (Tm5a/b or Dm9) in a loaf mutant also results in mistargeting.

a. If quantitative comparisons of the levels of Loaf are indeed relevant, then an essential experiment is to test whether increasing Loaf levels in the synaptic partners of R7 in a WT background cause mistargeting phenotypes.

b. Similarly, it is important to show that restoring Loaf in photoreceptors only in a loaf mutant background does not result in R7 mistargeting.

c. The authors do not distinguish between a 'matching' vs 'competition' model but data on when during development the mistargeting phenotype occurs may help clarify this- is it a mis-targeting phenotype or a maintenance phenotype?

2. The last section of the manuscript on the potential function of Loaf is largely speculative, though the authors are careful in their use of language.

*Reviewer #2:*

Our understanding of the molecular mechanisms that control circuit wiring processes remains limited. Using the *Drosophila* visual system as a model, Jessica Douthit and colleagues in Jessica Treisman's laboratory uncovered a key role for the CUB-LDLa protein Lost and found (Loaf) in regulating the targeting of R7 photoreceptor axons to their correct synaptic layer M6 in the medulla. The authors propose that Loaf, localized in endosomes, regulates the trafficking or function of cell surface molecules, such as the Lrp4 receptor, and, in this way, the formation of stable synapses. The genetic analysis of loaf requirements in adult R7 axons is detailed and thorough. The identification of a novel determinant mediating R7 axon targeting is significant and of wide interest for scientists studying neural circuit wiring. However, by focusing on the analysis of adult phenotypes, the core function of loaf remains unclear.

1. The authors use the term "restoring" loaf to neurons…, but is it indeed restoring? This would imply a removal and subsequent re-expression. Is it rather ectopic expression to increase levels, as only some experiments described rely on rescue approaches?

2. Loaf RNA interference (RNAi) mediated knockdown is described to lead to 30-70% of adult R7 axon mistargeting to the M3 layer. But what is the nature of this defect – is it a failure to stabilize axons followed by subsequent retraction? Or a failure to extend? It is crucial to assess phenotypes during developmental stages to determine as to which step is disrupted and when.

3. The authors test the hypothesis whether Loaf levels need to be matched between R7 axons and putative targets neurons in the medulla, focusing initially on R7 axons and their synaptic partners Dm8. However, this notion would only be justified, if one would know about expression levels of Loaf in pre- and postsynaptic neurons. Indeed, there does not seem to be any detailed information about Loaf expression levels in R7 neurons/cell bodies or axons. Moreover, the authors describe that Loaf is widely expressed in optic lobe neuron cell bodies in the 72 h APF pupal medulla. This seems late, considering that some cell surface molecules mediating targeting in the optic lobe cease expression from mid-pupal development onwards. It would therefore be important to examine expression at stages when defects begin to occur during development compared to controls. The provided image seems to show a gradient of cell body staining in a potentially damaged medulla cortex, and it is therefore not clear to what extent the antibody labeling reflects the real distribution of expression.

4. No defects could be detected when manipulating Loaf expression in specific neurons such as Dm8. However, when knockdown was achieved in large neuron populations such as glutamatergic or cholinergic neurons or those expression apt-Gal4, R7 axon targeting defects would occur. The authors then use this information to pinpoint the relevant partner neurons as Tm5a and b and Dm9. However, again it may seem a question of numbers rather than specificity. This would need to be addressed, possibly by taking into account how many neurons would be affected by each genetic manipulation.

5. The authors argue that R7 neurons match/compare its Loaf levels with multiple synaptic target neurons. But how would this be possible? It would be safer to state that levels may need to be similar, instead of implying an active matching process. Moreover, it is a concern that roles in medulla neurons have been assessed following over-expression of loaf, while reduction or loss do not seem to have an effect.

6. Loaf is localized in endosomes following over-expression. However, endogenous expression appears to be different. What is the evidence that the reporter – CG6024MI00316-GFSTF.1 is reflecting endogenous expression, and truly functioning as a protein trap? And could the authors test whether the ectopic protein was correctly expressed in neurons and their branches? Could the antibody help to get more insights?

7. The authors describe that Loaf regulates the activity of Lrp4. But is it indeed affecting activity or rather trafficking or correct spatio-temporal localization?

8. Defects for Lrp4 over-expression seem to be qualitatively different compared to Loaf related phenotypes. While there is some form of genetic interaction, it could be independent. It is also a concern that Loaf does not seem to affect expression of Lrp4 in line with the proposed role for Loaf. Moreover the "matching" hypothesis, would somehow require that loaf is regulating levels of the same or related molecules on the pre- and post-synaptic side.

9. The authors discuss and propose a role in synapse stabilization, but the study does not assess synapse formation, as the developmental emergence of phenotypes and requirements have not been addressed. The manuscript also alludes to a role in competition for a ligand or space, but this would require a deeper understanding of mutant phenotypes caused by manipulations on the pre- and postsynaptic side to support conclusions in this direction.

*Reviewer #3:*

The study by Treisman and colleagues addresses the developmental assembly of afferent axons into columnar circuit units, a highly conserved structural feature throughout nervous systems. From gene expression search, authors identified CUB-LDL cell surface protein (and named Loaf (Lost and found)) as a strong candidate for R7 photoreceptor axon guidance. Based on genetic manipulation, gene expression analysis, cell biological studies and cell type specific manipulations of the loaf gene, the authors could show that R7 axons terminate prematurely when the Loaf protein expression level is lower in R7 axons than in that of synaptic partner neurons. Author claims that Loaf localization is limited to intracellular vesicles where it may modulate the trafficking or modification of a transmembrane protein, and, in some way, the molecular system compares the levels of this protein on R7 and its synaptic partner whether or not to form a stable connection.

The manuscript is well written and the presented data are of high quality. The identification of novel players in axon circuit formation provides a new level of understanding about the complexity of the nervous system development. However, regarding the proposed gene function, the manuscript could be more convincing by addressing some issues regarding the function and the localization of Loaf protein.

1. While phenotypic analysis of adult R7 axons are shown, the sequence of developmental events are less clear. The difference between the 2 phenotypes is of importance: whether loaf mutant R7 stops prematurely or retract back from the synaptic layer. The author should show the loaf phenotype in pupal stages to demonstrate that the phenotype is premature stop and retraction. If it is a retraction, it suggests that the loaf function is the stabilization of R7 termini with the synaptic partner, and that the comparison of the Loaf protein is directly taking place at the synaptic sites. And again, if it is a retraction, I would like to see whether there is a strong genetic interaction with Lar hypomorphic mutant, where R7 are partially retracted from M6 layer to M3.

2. I am not 100% convinced from the current data that Loaf protein is not localised at the cell membrane. I must say that some of the transmembrane proteins appear to be strongly localized at internal membranes when transfected to S2 cells. I would like to suggest the authors to "surface label" the transfected S2 cells with antibodies of tagged protein without detergents. I also would like to see how the localization looks like when loaf transgene (UAS-loaf) is over-expressed in R7. The author showed whole-mount brain staining with anti-Loaf antibody (Figure 4B), but it seems that the antibody is trapped at the brain surface due to the high expression at the cortex. I would like to ask the authors to repeat this staining with longer incubation time (3-7 days) and/ or stronger detergents. Since the Mimic localization is shown, but I noticed some of the Mimic-GFP insertion lines cause mislocalization of the transmembrane protein (e.g localization at the cell body only) , I have to say that the Mimic-GFP localization is unreliable in this case.

[Editors’ note: further revisions were suggested prior to acceptance, as described below.]

Thank you for resubmitting your work entitled "R7 photoreceptor axon targeting requires matching levels of lost and found expression in R7 and its synaptic partners" for further consideration by *eLife*. Your revised article has been reviewed by 3 reviewers and the evaluation has been overseen by Utpal Banerjee as the Senior Editor, and a Reviewing Editor.

In your revised manuscript, you have examined the role of Lost and found (Loaf), a CUB-LDL domain protein in R7 axon targeting. You initially proposed a model to explain the context specific nature of whether Loaf LOF in photoreceptors causes an R7 mistargeting effect that involves comparisons of the levels of Loaf between R7s and (some) of its synaptic partners, resulting in appropriate R7 targeting when Loaf levels are matched. The reviewers all believe that your data are of high quality and provide a new level of understanding of circuit formation.

The manuscript has been improved but there are some remaining issues that need to be addressed, as outlined below:

Previously, the reviewers were not convinced that the model proposed was sufficiently supported by the data and asked for additional experiments. Now that you have successfully performed these experiments, you have found that the results don't support the original model and in the paper itself, while you qualify your text, it is not clear how loaf levels and matching actually works in this system.

As you state in your rebuttal letter and in the revised Discussion, that "R7 mistargeting results from the presence of Loaf in postsynaptic cells when it is absent in R7, and not necessarily from a quantitative comparison of relative Loaf levels in different cell types". And you further add: "Although the effect of Loaf on Lrp4 cannot fully explain its effect on R7, it at least provides proof of principle that Loaf could act by modulating the function of a cell surface protein." The reviewers and I believe that your paper indeed "makes a good start towards understanding the mechanism of action of this interesting molecule", but that "elucidating its mechanism is likely to be a long-term project."

That said, the reviewers and I in consultation believe that you could amend your very interesting paper primarily through textual changes, toward making the study appropriate for publication.

1. The reviewers did not feel that the model, though intriguing, is sufficiently supported by the data presented. The rebuttal letter is more in line with expressing (and arguing) the findings as they relate to your models and hypothesis, and thus we suggest you draw on you statements in the rebuttal letter that are clear and align with the current data. The Discussion would not need more detail, but it would help to remove complex models and hypotheses about the function of loaf in the Results, and recap a careful interpretation of the findings and a working model in the Discussion.

2. The developmental phenotype is not characterized in depth. You report a lack of defects early, but indeed the image showing this is a different stage than the control, and thus we do not know whether defects therefore represent a "retraction"/stabilization defect. We expect that it should be readily feasible to provide matching images for stages (and the authors might have them already) to strengthen the conclusion about developmental defects.

---

## [Author Response]

[Editors’ note: the authors resubmitted a revised version of the paper for consideration. What follows is the authors’ response to the first round of review.]

Reviewer #1:In this manuscript the authors focus on the role of Lost and found (Loaf), a CUB-LDL domain protein in R7 axon targeting. The context-specific nature of whether Loaf LOF in photoreceptors causes an R7 mistargeting effect is an interesting result and the authors have proposed a conceptually interesting model to explain the data, i.e. Quantitative comparisons of the levels of Loaf between R7s and their synaptic partners determine appropriate R7 targeting when Loaf levels are matched. However, while the data presented in this manuscript is consistent with the model, the model itself has not been tested sufficiently. Overall, I believe the work is interesting and has potential but requires essential additional data to support the central claims of the paper.1. The authors suggest that the R7 mistargeting phenotype is the consequence of lower loaf levels in R7s relative to R7 synaptic partners (Tm5a/b and/or Dm9) since removing Loaf from photoreceptors alone results in mistargeting and likewise restoring Loaf in synaptic partners of R7 (Tm5a/b or Dm9) in a loaf mutant also results in mistargeting.a. If quantitative comparisons of the levels of Loaf are indeed relevant, then an essential experiment is to test whether increasing Loaf levels in the synaptic partners of R7 in a WT background cause mistargeting phenotypes.

The reviewer wanted us to test whether overexpression of Loaf in the synaptic partners of R7 in a wild-type context caused R7 mistargeting. We have tried overexpressing Loaf using the same GAL4 drivers that produced phenotypes in the rescue experiment: *GMR9D03-GAL4, dve-GAL4 + vg-GAL4*, and *ap-GAL4 + tj-GAL4*, but have not observed targeting defects (Figure 5—figure supplement 2). This may be due to the limited sensitivity of our assay, as the mistargeting we observe when Loaf is misexpressed in a *loaf* mutant background is significant but not severe. It might therefore be difficult for us to detect the effect of a more subtle difference in expression levels. In addition, we now observe that Loaf is normally enriched in R7 terminals, raising the possibility that the overexpression we are able to achieve with these GAL4 lines does not exceed the normal level in R7. In light of these uncertainties, we have revised the Discussion to emphasize that R7 mistargeting results from the presence of Loaf in postsynaptic cells when it is absent in R7, and not necessarily from a quantitative comparison of relative Loaf levels in different cell types.

b. Similarly, it is important to show that restoring Loaf in photoreceptors only in a loaf mutant background does not result in R7 mistargeting.

The reviewer asked us to show that restoring Loaf only in photoreceptors in a *loaf* mutant background does not cause R7 mistargeting. We have added these data to Figure 4G.

c. The authors do not distinguish between a 'matching' vs 'competition' model but data on when during development the mistargeting phenotype occurs may help clarify this- is it a mis-targeting phenotype or a maintenance phenotype?

The reviewer asked when in development the mistargeting occurs. We have examined 40h APF and 60h APF pupae in which *loaf* is knocked down in photoreceptors. The first stage of R7 targeting to its temporary layer at 40h appears normal, but there is significant mistargeting at 60h. This suggests that the defect occurs when the initial contacts of R7 and its target cells are first being formed and stabilized, consistent with our matching hypothesis. We have added these data as Figure 1G-K.

2. The last section of the manuscript on the potential function of Loaf is largely speculative, though the authors are careful in their use of language.

The reviewer found the section on the potential function of Loaf largely speculative, but did not suggest a way for us to remedy this. Determining its mechanism of action is technically very challenging, considering its intracellular localization, the absence of information about potential partners, and the difficulty of expressing genes in specific cell types early enough for them to affect synaptogenesis. Although the effect of Loaf on Lrp4 cannot fully explain its effect on R7, it at least provides proof of principle that Loaf could act by modulating the function of a cell surface protein. We think that our paper makes a good start towards understanding the mechanism of action of this interesting molecule, and fully elucidating its mechanism is likely to be a long-term project.

Reviewer #2:Our understanding of the molecular mechanisms that control circuit wiring processes remains limited. Using the *Drosophila* visual system as a model, Jessica Douthit and colleagues in Jessica Treisman's laboratory uncovered a key role for the CUB-LDLa protein Lost and found (Loaf) in regulating the targeting of R7 photoreceptor axons to their correct synaptic layer M6 in the medulla. The authors propose that Loaf, localized in endosomes, regulates the trafficking or function of cell surface molecules, such as the Ldl4 receptor, and, in this way, the formation of stable synapses. The genetic analysis of loaf requirements in adult R7 axons is detailed and thorough. The identification of a novel determinant mediating R7 axon targeting is significant and of wide interest for scientists studying neural circuit wiring. However, by focusing on the analysis of adult phenotypes, the core function of loaf remains unclear.1. The authors use the term "restoring" loaf to neurons…, but is it indeed restoring? This would imply a removal and subsequent re-expression. Is it rather ectopic expression to increase levels, as only some experiments described rely on rescue approaches?

The reviewer questioned our use of the term “restoring” Loaf function. We only use this term for experiments in which we supply Loaf to specific cell types in a *loaf* mutant background, which the reviewer appears to agree is an appropriate use of the word.

2. Loaf RNA interference (RNAi) mediated knockdown is described to lead to 30-70% of adult R7 axon mistargeting to the M3 layer. But what is the nature of this defect – is it a failure to stabilize axons followed by subsequent retraction? Or a failure to extend? It is crucial to assess phenotypes during developmental stages to determine as to which step is disrupted and when.

The reviewer thought that we should look at the effects of loss of *loaf* earlier in development to determine whether the phenotype is a failure of axonal extension or a later retraction. As described in the response to reviewer 1 point 1c, we have now shown that the phenotype arises during the second stage of R7 targeting (Figure 1G-K). Other studies have shown that there is no active axonal extension during the second stage, and phenotypes that arise at this time, as in *Lar* mutants, have been interpreted as a failure to form stable synaptic contacts. This results in R7 terminals becoming separated from their target dendrites as these are displaced down to the M6 layer (Ozel et al., 2015, 2019).

3. The authors test the hypothesis whether Loaf levels need to be matched between R7 axons and putative targets neurons in the medulla, focusing initially on R7 axons and their synaptic partners Dm8. However, this notion would only be justified, if one would know about expression levels of Loaf in pre- and postsynaptic neurons. Indeed, there does not seem to be any detailed information about Loaf expression levels in R7 neurons/cell bodies or axons. Moreover, the authors describe that Loaf is widely expressed in optic lobe neuron cell bodies in the 72 h APF pupal medulla. This seems late, considering that some cell surface molecules mediating targeting in the optic lobe cease expression from mid-pupal development onwards. It would therefore be important to examine expression at stages when defects begin to occur during development compared to controls. The provided image seems to show a gradient of cell body staining in a potentially damaged medulla cortex, and it is therefore not clear to what extent the antibody labeling reflects the real distribution of expression.

The reviewer wanted us to provide more information about Loaf expression in R7 and in medulla neurons. This is quite difficult to do, because of the widespread expression of Loaf in the optic lobes and the cross-reactivity of our antibody with a protein present in cell bodies. This was our second attempt to make an antibody; the first did not work at all. The reviewer also asked us to look at Loaf expression in the medulla at earlier pupal stages. We have now found that in 40h and 60h APF pupae, Loaf is enriched in R7 axons and terminals in the medulla, and that this enrichment is lost when *loaf* is knocked down in photoreceptors. We have replaced the original Figure 4B, C with these data, which are shown in Figure 4A-C and Figure 4—figure supplement 1C, D.

4. No defects could be detected when manipulating Loaf expression in specific neurons such as Dm8. However, when knockdown was achieved in large neuron populations such as glutamatergic or cholinergic neurons or those expression apt-Gal4, R7 axon targeting defects would occur. The authors then use this information to pinpoint the relevant partner neurons as Tm5a and b and Dm9. However, again it may seem a question of numbers rather than specificity. This would need to be addressed, possibly by taking into account how many neurons would be affected by each genetic manipulation.

The reviewer thought that the effects of *loaf* restoration with different GAL4 drivers might depend on the number of cells expressing the driver rather than on their identity. Although we cannot fully rule this out, *GMR9D03-GAL4* has a strong effect despite being expressed in relatively few cells (Figure 5—figure supplement 1A, B), and *dve-GAL4* also appears quite specific to Dm9 (Figure 5—figure supplement 1C).

5. The authors argue that R7 neurons match/compare its Loaf levels with multiple synaptic target neurons. But how would this be possible? It would be safer to state that levels may need to be similar, instead of implying an active matching process. Moreover, it is a concern that roles in medulla neurons have been assessed following over-expression of loaf, while reduction or loss do not seem to have an effect.

The reviewer thought we were overstating our results by saying that R7 must match its Loaf levels to its target neurons, since loss of *loaf* from medulla neurons does not have an effect on R7. We have now emphasized the asymmetry of this matching process; targeting is abnormal when R7 has less Loaf than its partner neurons, but not vice versa. It is possible that an excess of Loaf in the synaptic partners prevents the contact from being stabilized.

6. Loaf is localized in endosomes following over-expression. However, endogenous expression appears to be different. What is the evidence that the reporter – CG6024MI00316-GFSTF.1 is reflecting endogenous expression, and truly functioning as a protein trap? And could the authors test whether the ectopic protein was correctly expressed in neurons and their branches? Could the antibody help to get more insights?

The reviewer was concerned about the difference in localization between overexpressed Loaf and the tagged endogenous protein. Our improved antibody staining has enabled us to show that there is indeed a difference; the tagged protein shows less localization to the neuropil and no visible enrichment in R7 terminals (Figure 4—figure supplement 1A). We believe that the tag disrupts the localization and function of the protein, as clones homozygous for the MiMIC insertion show R7 mistargeting (Figure 4—figure supplement 1B).

7. The authors describe that Loaf regulates the activity of Lrp4. But is it indeed affecting activity or rather trafficking or correct spatio-temporal localization?

The reviewer asked whether Loaf could be affecting the trafficking or localization of Lrp4 rather than its activity. We meant to use the term “activity” broadly to include all of these possibilities. We have now clarified this in the text. However, we have not been able to detect an effect of *loaf* on Lrp4 localization to photoreceptor axons (Figure 7—figure supplement 2E, G).

8. Defects for Lrp4 over-expression seem to be qualitatively different compared to Loaf related phenotypes. While there is some form of genetic interaction, it could be independent. It is also a concern that Loaf does not seem to affect expression of Lrp4 in line with the proposed role for Loaf. Moreover the "matching" hypothesis, would somehow require that loaf is regulating levels of the same or related molecules on the pre- and post-synaptic side.

The reviewer points out that the effects of Loaf on Lrp4 would not explain the R7 phenotype caused by *loaf*, because Lrp4 is only on one side of the synapse and has distinct phenotypes. We did not mean to imply that Lrp4 is the only Loaf-dependent protein relevant for the R7 phenotype; we agree that this is unlikely, since we show that *loaf* knockdown still disrupts R7 targeting in an *Lrp4* mutant background (Figure 7—figure supplement 2J, K). We simply used it as an example to show that Loaf can affect the function of a cell surface protein involved in synapse organization. We have now clarified this in the text.

9. The authors discuss and propose a role in synapse stabilization, but the study does not assess synapse formation, as the developmental emergence of phenotypes and requirements have not been addressed. The manuscript also alludes to a role in competition for a ligand or space, but this would require a deeper understanding of mutant phenotypes caused by manipulations on the pre- and postsynaptic side to support conclusions in this direction.

The reviewer pointed out that we had not addressed synapse formation or the developmental emergence of the phenotype. We have now looked at when the phenotype arises (see point 2) and have removed the language about synaptogenesis. It is difficult to look at synapse formation directly, since simply assessing the distribution of some synaptic markers would not show that the synapse is functional.

Reviewer #3:[…] The manuscript is well written and the presented data are of high quality. The identification of novel players in axon circuit formation provides a new level of understanding about the complexity of the nervous system development. However, regarding the proposed gene function, the manuscript could be more convincing by addressing some issues regarding the function and the localization of Loaf protein.1. While phenotypic analysis of adult R7 axons are shown, the sequence of developmental events are less clear. The difference between the 2 phenotypes is of importance: whether loaf mutant R7 stops prematurely or retract back from the synaptic layer. The author should show the loaf phenotype in pupal stages to demonstrate that the phenotype is premature stop and retraction. If it is a retraction, it suggests that the loaf function is the stabilization of R7 termini with the synaptic partner, and that the comparison of the Loaf protein is directly taking place at the synaptic sites. And again, if it is a retraction, I would like to see whether there is a strong genetic interaction with Lar hypomorphic mutant, where R7 are partially retracted from M6 layer to M3.

The reviewer asked us to look at the timing of the mistargeting caused by loss of *loaf* to determine whether the phenotype is a premature stop or a retraction. As described in the response to reviewer 1 point 1c, we have now shown that the phenotype arises between the first and second stages of R7 targeting (Figure 1G-K). Other studies have shown that there is no active axonal extension during the second stage, and phenotypes that arise during this time, as in *Lar* mutants, have been interpreted as a failure to stabilize synaptic contacts. This results in R7 terminals becoming separated from their target dendrites as these are displaced down to the M6 layer (Ozel et al., 2015, 2019). The reviewer wanted us to look for a genetic interaction with *Lar* hypomorphic mutants. Since the *loaf* phenotype can only be observed with photoreceptor-specific RNAi or in clones, we have done this experiment by combining *loaf* RNAi with *Lar* RNAi. We find that with the *lGMR-GAL4* driver, *loaf* RNAi has a weak effect on R7 and *Lar* RNAi an intermediate one, but combining the two produces almost complete R7 mistargeting (Figure 7A-D). This synergistic effect suggests that both genes contribute to the same process. Consistent with this conclusion, we find that *loaf* knockdown also increases R7 mistargeting in *Liprin-α* mutants, although this effect could simply be additive (Figure 7—figure supplement 1G-J).

2. I am not 100% convinced from the current data that Loaf protein is not localised at the cell membrane. I must say that some of the transmembrane proteins appear to be strongly localized at internal membranes when transfected to S2 cells. I would like to suggest the authors to "surface label" the transfected S2 cells with antibodies of tagged protein without detergents. I also would like to see how the localization looks like when loaf transgene (UAS-loaf) is over-expressed in R7. The author showed whole-mount brain staining with anti-Loaf antibody (Figure 4B), but it seems that the antibody is trapped at the brain surface due to the high expression at the cortex. I would like to ask the authors to repeat this staining with longer incubation time (3-7 days) and/ or stronger detergents. Since the Mimic localization is shown, but I noticed some of the Mimic-GFP insertion lines cause mislocalization of the transmembrane protein (e.g localization at the cell body only), I have to say that the Mimic-GFP localization is unreliable in this case.

The reviewer thought we should investigate whether some Loaf is present on the plasma membrane by surface labeling of transfected cells. Using a V5 tag at the N-terminus of Loaf to look at surface expression in S2 cells, we have found that Loaf cannot be detected on unpermeabilized cells, although another extracellularly tagged cell surface protein, Sdk, is detectable (Figure 6H, I). He or she also suggested staining pupal brains with a longer incubation time and stronger detergents. We thank the reviewer for these helpful suggestions. Our improved pupal brain staining showed that in 40h APF and 60 h APF pupae, Loaf is enriched in R7 axons and terminals in the medulla, and this enrichment is lost when *loaf* is knocked down in photoreceptors. We have replaced the original Figure 4B, C with these data, which are shown in Figure 4A-C and Figure 4—figure supplement 1C, D. The reviewer also asked to see the localization of Loaf when it is overexpressed in R7. We find that HA-Loaf expressed in clones of photoreceptors is transported to the axons and terminals of R7 and R8 (Figure 4—figure supplement 1F).

[Editors’ note: what follows is the authors’ response to the second round of review.]

[…] The manuscript has been improved but there are some remaining issues that need to be addressed, as outlined below:Previously, the reviewers were not convinced that the model proposed was sufficiently supported by the data and asked for additional experiments. Now that you have successfully performed these experiments, you have found that the results don't support the original model and in the paper itself, while you qualify your text, it is not clear how loaf levels and matching actually works in this system.As you state in your rebuttal letter and in the revised Discussion, that "R7 mistargeting results from the presence of Loaf in postsynaptic cells when it is absent in R7, and not necessarily from a quantitative comparison of relative Loaf levels in different cell types". And you further add: "Although the effect of Loaf on Lrp4 cannot fully explain its effect on R7, it at least provides proof of principle that Loaf could act by modulating the function of a cell surface protein." The reviewers and I believe that your paper indeed "makes a good start towards understanding the mechanism of action of this interesting molecule", but that "elucidating its mechanism is likely to be a long-term project."That said, the reviewers and I in consultation believe that you could amend your very interesting paper primarily through textual changes, toward making the study appropriate for publication.1. The reviewers did not feel that the model, though intriguing, is sufficiently supported by the data presented. The rebuttal letter is more in line with expressing (and arguing) the findings as they relate to your models and hypothesis, and thus we suggest you draw on you statements in the rebuttal letter that are clear and align with the current data. The Discussion would not need more detail, but it would help to remove complex models and hypotheses about the function of loaf in the Results, and recap a careful interpretation of the findings and a working model in the Discussion.

The reviewers asked us to rewrite the paper to remove the more complex models and bring the interpretations of our results more in line with the way we stated them in the rebuttal letter. We have done this by removing our discussion of the competition model and the diagram of this model in Figure 4, in order to focus on the importance of the relative levels of Loaf in R7 and its synaptic partners. We have also made it clear in the Abstract, Introduction, Results and Discussion sections that we have only shown that the presence of Loaf in synaptic partner neurons when it is absent in R7 is detrimental to R7 targeting, and not that the levels of Loaf in R7 and its synaptic partners need to be precisely matched. We have kept the diagrams in Figures 3 and 5 and included additional panels in Figure 5 to make it a more comprehensive working model, because we think these diagrams will help readers to understand the rationale for and interpretation of the experiments we performed.

2. The developmental phenotype is not characterized in depth. You report a lack of defects early, but indeed the image showing this is a different stage than the control, and thus we do not know whether defects therefore represent a "retraction"/stabilization defect. We expect that it should be readily feasible to provide matching images for stages (and the authors might have them already) to strengthen the conclusion about developmental defects.

The reviewers pointed out that Figure 1I showed *GMR-GAL4>loaf RNAi* at a later stage of pupal development than the control in Figure 1H. We apologize for this oversight. We had intended to replace these images with better stage-matched examples, but somehow neglected to do so. We have now replaced them with images that more clearly show normal R7 targeting at 40 h APF when *loaf* is knocked down in photoreceptors, indicating that the defect arises at the second stage when synaptic contacts are stabilized and the R7 terminal passively moves down to the M6 layer.